# DR²Seg: Decomposed Two-Stage Rollouts for Efficient Reasoning Segmentation in Multimodal Large Language Models

**Yulin He\*** [1]   **Wei Chen\*** [1]   **Zhikang Jian** [1]   **Tianhang Guo** [1]   **Wenjuan Zhou** [1]   **Minglong Li** [1]
**Shaowu Yang** [1]   **Wenjing Yang** [1]

## Abstract

Reasoning segmentation is an emerging vision-language task that requires reasoning over intricate text queries to precisely segment objects. However, existing methods typically suffer from overthinking, generating verbose reasoning chains that interfere with object localization in multimodal large language models (MLLMs). To address this issue, we propose DR²Seg, a self-rewarding framework that improves both reasoning efficiency and segmentation accuracy without requiring extra thinking supervision. DR²Seg employs a two-stage rollout strategy that decomposes reasoning segmentation into multimodal reasoning and referring segmentation. In the first stage, the model generates a self-contained description that explicitly specifies the target object. In the second stage, this description replaces the original complex query to verify its self-containment. Based on this design, two self-rewards are introduced to mitigate overthinking and the associated attention dispersion. Extensive experiments conducted on 3B and 7B variants of Qwen2.5-VL, as well as on both SAM2 and SAM3, demonstrate that DR²Seg consistently improves reasoning efficiency and overall segmentation accuracy. The source code can be found at https://github.com/harrylin-hyl/DR2Seg.

## 1. Introduction

Multimodal large language models (MLLMs) (Liu et al., 2023b; Bai et al., 2025; OpenAI., 2024) encode rich open-world knowledge and exhibit strong capabilities in joint image-text understanding across downstream tasks. By

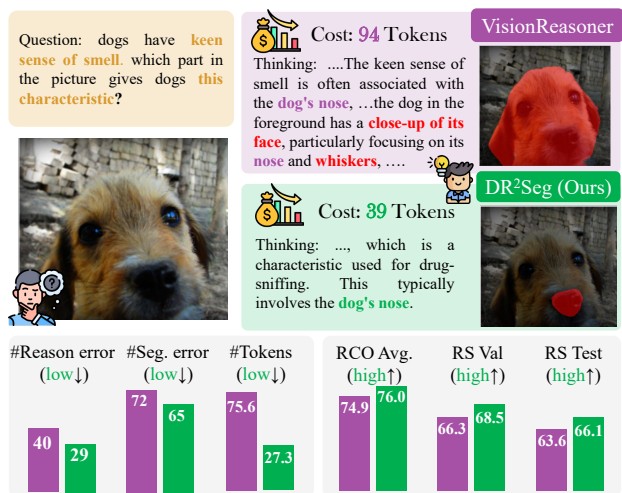

*Figure 1.* **Motivation of DR²Seg.** Verbose reasoning can mislead MLLMs to localize false regions (e.g., face or whiskers) instead of the true target (nose). DR²Seg mitigates such reasoning and localization errors, achieving efficient reasoning (∼3× shorter) and accurate segmentation on RefCOCO (RCO) and ReasonSeg (RS).

leveraging these strengths, MLLMs can analyze visual scenes and interpret ambiguous human intents, enabling more complex tasks. Recently, reasoning segmentation (Yu et al., 2016; Lai et al., 2024; Zhu et al., 2025) which builds upon MLLMs and emphasizes the synergy between reasoning and perception, has garnered widespread attention. Its core objective is to segment target objects from complex textual queries. Compared to the referring segmentation task (Ding et al., 2021; Yang et al., 2022; Liu et al., 2023c), reasoning segmentation involves more intricate and implicit queries, making it more challenging and more suitable for real-world agent scenarios, such as interactive robotics (Yin et al., 2023) and autonomous driving (Tian et al., 2024).

Currently, there are two main paradigms in reasoning segmentation. Supervised fine-tuning (SFT)-based methods (Lai et al., 2024; Ren et al., 2024), which integrate pre-trained MLLMs with segmentation models through SFT to enable reasoning segmentation. However, SFT-based methods exhibit limited generalization to out-of-distribution (OOD) scenarios and lack explicit reasoning chains for ex-

---

\*Equal contribution [1]College of Computer Science and Technology, National University of Defense Technology, Changsha, China. Correspondence to: Wei Chen <chenwei@nudt.edu.cn>.

*Proceedings of the 43rd International Conference on Machine Learning*, Seoul, South Korea. PMLR 306, 2026. Copyright 2026 by the author(s).

plainability (Liu et al., 2025a; Wang et al., 2025). In contrast, reinforcement learning (RL)-based methods (Liu et al., 2025b; Wang et al., 2025), inspired by ZeroSeg (Liu et al., 2025a), optimize MLLMs with perception-oriented rewards using GRPO (Shao et al., 2024). By adaptively generating reasoning chains, these methods can achieve improved OOD generalization. However, existing RL-based methods often suffer from overthinking, generating verbose reasoning chains that not only reduce computational efficiency but also interfere with accurate object localization. To address this issue, PixelThink (Wang et al., 2025) leverages an extra large-scale expert MLLM (i.e., Qwen2.5-VL-72B (Bai et al., 2025)) to estimate problem difficulty, thereby implicitly introducing thinking supervision. However, such a strategy depends strongly on the expert MLLM as an auxiliary module, without probing into the intrinsic self-organizing reasoning capacities of the base model itself.

These issues motivate us to explore self-rewarding (Yuan et al., 2024; Zhou et al., 2024), a recent advancement in reasoning MLLMs that remains underexplored in reasoning perception. Unlike visual question answering in reasoning MLLMs, reasoning perception is more prone to attention confusion due to inherent modality differences between coordinates and text. To address this challenge, we propose DR$^2$Seg, which adopts a two-stage rollout strategy to decompose reasoning segmentation into multimodal reasoning and referring segmentation. Training involves two rollout passes of the same MLLM: 1) In the first pass, the model generates an explicit inferring description to specify target objects. 2) In the second pass, it is re-prompted to respond based on this description. If the second-pass prediction is correct, the description is regarded as faithful and receives a self-reward. Furthermore, we introduce a length-based self-reward that encourages concise reasoning under explicit description guidance, thereby reducing unnecessary thinking tokens. As a result, DR$^2$Seg achieves efficient reasoning and accurate localization without thinking supervision, as shown in Fig. 1. Our contributions are summarized as:

- We propose DR$^2$Seg, a simple yet effective self-reward framework that enhances both efficiency and segmentation accuracy using only the model's intrinsic capability, without requiring extra MLLMs or supervision.

- DR$^2$Seg designs a two-stage rollout strategy that decouples multimodal reasoning and perception in MLLM for accurate segmentation, combined with a length-based self-reward to reduce redundant reasoning.

- Extensive experiments validate the effectiveness and generalization of DR$^2$Seg across MLLMs of varying scales and segmentation models, offering valuable insights into efficient reasoning perception.

## 2. Related Work

### 2.1. Reasoning Segmentation

Image segmentation has evolved from the traditional closed-set setting (Ronneberger et al., 2015; Minaee et al., 2021) to open-vocabulary setting of referring segmentation (Ding et al., 2021; Yang et al., 2022; Liu et al., 2023c), where objects are segmented using brief descriptions. Recently, benefiting from the visual-language reasoning capabilities of MLLMs, reasoning segmentation (Shen et al., 2025) has further expanded prompts from fixed vocabularies to arbitrary linguistic forms.

The pioneering work LISA (Shen et al., 2025) integrates MLLMs with the segmentation model SAM (Kirillov et al., 2023) by aligning textual reasoning with segmentation. Building on LISA, a series of studies explore supervised fine-tuning to strengthen the alignment between textual tokens and fine-grained segmentation (Yang et al., 2023; Ren et al., 2024). However, SFT-based methods suffer from limited generalization, leading to notable performance degradation in OOD scenarios. Seg-Zero (Liu et al., 2025a) addresses this issue by introducing an reinforcement learning based framework, which leverages GRPO (Shao et al., 2024) to adaptively optimize the model's reasoning ability, achieving improved generalization. VisionReasoner (Liu et al., 2025b) further extends to enable multi-object segmentation by incorporating a bipartite matching algorithm. PixelThink (Wang et al., 2025) focuses on the efficiency and introduces an auxiliary large-scale MLLM to estimate query difficulty, thereby improving reasoning efficiency. In contrast, this paper proposes a self-reward framework that requires no additional MLLMs while achieving more efficient and accurate performance.

### 2.2. Self-Rewarding Reinforcement Learning

High-quality rewards are critical for reinforcement learning with verifiable rewards (RLVR), which typically relies on high-quality reward models or even human feedback, becoming a major bottleneck for scalability (Peng et al., 2025; Wen et al., 2025; Su et al., 2025). To address this dilemma, recent works have explored self-rewarding approaches, where reward signals are derived from the model itself (Yuan et al., 2024). In language-model-based settings, self-rewarding methods replace external reward models with signals such as model confidence (Li et al., 2025a; van Niekerk et al., 2025), uncertainty (Zhao et al., 2025), or self-verified solutions (Simonds et al., 2025).

Recently, several studies have extended this paradigm to MLLMs. For example, Calibrated Self-Rewarding (Zhou et al., 2024) iteratively generates responses and performs self-scoring, assigning rewards through progressively applied visual constraints. PLARE (Luu et al., 2025) queries

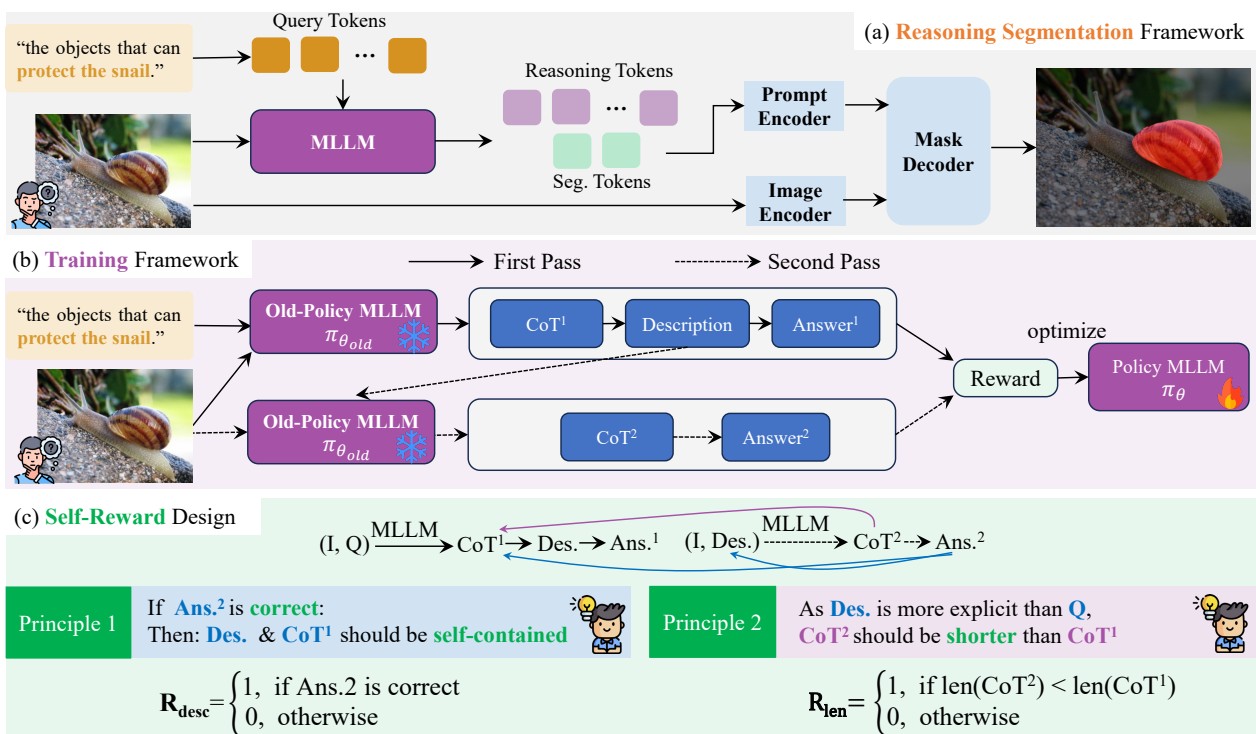

*Figure 2.* **Overview of DR²Seg.** (a) DR²Seg performs a two-stage rollout. In this first pass, the model takes an image-query pair and produces a structured output comprising a CoT, a description, and an answer. In the second pass, the model is re-prompted with the image and the generated description, replacing the original query. (b) DR²Seg adopts a self-reward mechanism to optimize the MLLM, enabling more efficient reasoning and accurate segmentation.

a MLLM to obtain preference labels over pairs of visual trajectory segments, and directly trains the policy with a supervised contrastive preference learning, eliminating the need for an explicit reward model. Vision-SR1 (Li et al., 2025b) decomposes MLLM reasoning into visual perception and language reasoning by explicitly rewarding visual perception. This work further explores self-rewarding for reasoning perception, focusing on instance-level object understanding through decomposition into reasoning and referring segmentation via an explicit description bottleneck. Unlike most prior methods that rely on textual descriptions for Visual Question Answering (VQA), reasoning segmentation relies more heavily on fine-grained visual details, making models more vulnerable to attention distraction caused by redundant or irrelevant preceding reasoning steps.

## 3. Methodology

### 3.1. Problem Definition

Given an input image $\mathcal{I}$ and an implicit textual query $\mathcal{Q}$, reasoning segmentation (Lai et al., 2024) aims to produce a binary segmentation mask $\mathcal{M}$. This task is similar to referring segmentation (Ding et al., 2021) but is more challenging, as reasoning segmentation involves more complex queries expressed in arbitrary free-form natural language.

Moreover, reasoning segmentation also emphasizes the generation of reasoning chains $\mathcal{R}$, which play a crucial role in understanding intent and reasoning to identify targets, thereby improving explainability and generalization.

### 3.2. Overview

We follow the standard reasoning segmentation framework (Liu et al., 2025b) as shown in Fig. 2(a), which includes a reasoning MLLM and a segmentation model. The MLLM takes an image $\mathcal{I}$ and a textual query $\mathcal{Q}$ as input, and produces two outputs: $\mathcal{R}, \mathcal{A} = \text{MLLM}(\mathcal{I}, \mathcal{Q})$. Here, $\mathcal{R}$ denotes reasoning chains of the MLLM and $\mathcal{A}$ represents spatial answers, including a bounding box, a point, and an optional description, which serve as inputs of the segmentation model. We adopt the SAM series (Kirillov et al., 2023; Ravi et al.; Carion et al., 2025) as the segmentation models, which take the image $\mathcal{I}$ and the answers $\mathcal{A}$ as input and generate the binary masks $\mathcal{M}$ for target objects.

### 3.3. DR²Seg: A Pure Self-Reward Framework

As discussed, redundant over-thinking in MLLMs confuses subsequent localization and degrades both efficiency and accuracy. However, supervising the reasoning process is inherently challenging because multiple reasoning paths can

lead to correct answers, which largely explains the limited generalization of SFT-based methods. PixelThink (Wang et al., 2025) addresses this issue by introducing an expert MLLM to constrain reasoning length. However, this implicitly injects external knowledge from a larger model (i.e., Qwen2.5VL-72B), raising fairness concerns and hindering the model's ability to self-evolve. Instead, we propose a self-reward framework consisting of two-stage rollout and self-reward design, as shown in Fig. 2(b) and Fig. 2(c).

**Two-Stage Rollout.** To encourage MLLMs to produce self-contained reasoning for segmentation, we enforce a "think then description" generation format. Given an image $\mathcal{I}$ and a textual query $\mathcal{Q}$, the MLLM outputs a structured response as: <think> $\mathcal{R}$ </think><description> $\mathcal{D}$ </description> <answer> $\mathcal{A}$ </answer>, where $\mathcal{D}$ denotes inferring descriptions.

The training involves two rollout passes of the same MLLM:

(1) First pass: $(\mathcal{I}, \mathcal{Q}) \rightarrow (\mathcal{R}^1, \mathcal{D}, \mathcal{A}^1)$, where the model generates an explicit inferring description to specify the target objects.

(2) Second pass: $(\mathcal{I}, \mathcal{D}) \rightarrow (\mathcal{R}^2, \mathcal{A}^2)$, in which the model is re-prompted to reason based on the explicit description.

During training, the first pass performs multimodal reasoning to generate referring descriptions, and the second pass generates spatial answers based on these descriptions, decomposing reasoning segmentation into multimodal reasoning and referring segmentation. Notably, only a single pass (i.e., the first pass) is required during inference, thereby maintaining computational efficiency.

**Self-Reward Design.** We can combine the two rollout passes into a longer reasoning path based on the token generation order of the MLLM, formalized as: $(\mathcal{I}, \mathcal{Q}) \rightarrow \mathcal{R}^1 \rightarrow \mathcal{D} \rightarrow (\mathcal{I}, \mathcal{D}) \rightarrow \mathcal{R}^2 \rightarrow \mathcal{A}^2$. Based on this reasoning path, we infer whether the preceding input is self-contained according to the correctness of the answer. We then derive two guiding principles and design corresponding self-rewards:

*Principle 1: If $\mathcal{A}^2$ is correct, then $\mathcal{D}$ and $\mathcal{R}^1$ should be self-contained.*

Accurate segmentation heavily relies on language descriptions, given the diverse number and granularity of objects in the image. Therefore, the answer from the second rollout pass serves to verify the reasoning and informational completeness of the first-pass generation. If the model can still produce the correct answer given only $(I, \mathcal{D})$, we consider $\mathcal{D}$ to be correct and faithful, and accordingly assign a description self-reward $\mathbf{R}_{\text{desc}}$:

$$\mathcal{R}^2, \mathcal{A}^2 = \text{MLLM}(\mathcal{I}, \mathcal{D}), \quad (1)$$

$$\mathbf{R}_{\text{desc}}(\mathcal{A}^2, \mathcal{A}^*) = \mathbf{R}_{\text{acc}}(\mathcal{A}^2, \mathcal{A}^*), \quad (2)$$

where $\mathcal{A}^*$ is the ground-truth answer and $\mathbf{R}_{\text{acc}}$ is the answer

accuracy reward as in (Liu et al., 2025b).

*Principle 2: As $\mathcal{D}$ is more semantic explicit than $\mathcal{Q}$, $\mathcal{R}^2$ should be shorter than $\mathcal{R}^1$.*

After multimodal reasoning in the first pass, $\mathcal{D}$ provides a concise phrase-level description of the target, which should lead to shorter reasoning chains $\mathcal{R}^2$ compared to those in the first pass $\mathcal{R}^1$. We first compute the token number of $\mathcal{R}^1$ and $\mathcal{R}^2$:

$$\mathcal{N}^1 = \text{len}(\text{Token}(\mathcal{R}^1)), \quad (3)$$

$$\mathcal{N}^2 = \text{len}(\text{Token}(\mathcal{R}^2)), \quad (4)$$

where Token($\cdot$) denotes the tokenizer of the MLLM, and len($\cdot$) returns the length of the tokens. Then, we define the length-based self-reward:

$$\mathbf{R}_{\text{len}} = \text{clip}\big(\mathbb{I}\big[\mathcal{N}^2 < \mathcal{N}^1\big] - \gamma \max\big(0, \mathcal{N}^1 - \mathcal{N}_0\big), 0, 1\big). \quad (5)$$

Here, the first term $\mathbb{I}\big[\mathcal{N}^2 < \mathcal{N}^1\big]$ follows Principle 2 by comparing the reasoning lengths of the two passes, thereby promoting more concise and refined reasoning. However, since the first term is a purely comparative reward, using it alone lacks an absolute anchor and allows the model to exploit reward hacking: $\mathcal{N}^1$ and $\mathcal{N}^2$ increase synchronously. Therefore, we introduce the second term: $\gamma \max\big(0, \mathcal{N}^1 - \mathcal{N}_0\big)$, where $\mathcal{N}_0$ is a predefined length anchor and $\gamma$ controls the strength of the length penalty. Finally, we apply a clip($\cdot$) operator to ensure that $\mathbf{R}_{\text{len}}$ remains within the range [0,1].

Instead of relying on an external reward model (e.g., Qwen2.5VL-72B in PixelThink (Wang et al., 2025)), we leverage the model's own capabilities for self-evaluation. By verifying the correctness of the second-pass answer and comparing the reasoning lengths across the two passes, our method enables self-rewarding over the reasoning process.

**Total Reward.** First, we adopt the base reward $\mathbf{R}_{\text{base}}$ from the baseline VisionReasoner (Liu et al., 2025b), which consists of a format reward, a non-repeat reward, and a answer accuracy reward:

$$\mathbf{R}_{\text{base}} = \mathbf{R}_{\text{format}} + \mathbf{R}_{\text{non-repeat}} + \mathbf{R}_{\text{acc}}. \quad (6)$$

These rewards encourage structured output, discourage repetitive responses, and ensure accurate segmentation.

Then, we incorporate the proposed description self-reward $\mathbf{R}_{\text{desc}}$ and length-based self-reward $\mathbf{R}_{\text{len}}$ into the overall reward. The total reward is computed as:

$$\mathbf{R}_{\text{total}} = \big(\mathbf{R}_{\text{base}} + \mathbf{R}_{\text{desc}}\big) \cdot \tilde{\mathbf{R}}_{\text{len}}, \quad (7)$$

$\mathbf{R}_{\text{desc}}$ evaluates the correctness of the answer generated in the second rollout pass. $\tilde{\mathbf{R}}_{\text{len}}$ denotes the conditional length-based self-reward, defined as:

$$\tilde{\mathbf{R}}_{\text{len}} = \begin{cases} \mathbf{R}_{\text{len}}, & \text{if } \exists i \in \{1, \ldots, n\}, \mathbf{R}_{\text{acc}}^{(i)} > 0, \\ 1, & \text{otherwise.} \end{cases} \quad (8)$$

When all $n$ rollouts yield zero accuracy reward, $\tilde{\mathbf{R}}_{\text{len}}$ is disabled by setting to 1. This avoids imposing premature length constraints before successful target localization and encourages more active exploration in the early stage. In this way, the model can dynamically balance reasoning length against answer accuracy based on problem difficulty. To ensure training stability, we integrate $\tilde{\mathbf{R}}_{\text{len}}$ into the reward function using a multiplicative formulation.

**Training with GRPO.** We adopt Group-Relative Policy Optimization (GRPO) (Shao et al., 2024) to fine-tune the model, maximizing the total reward $\mathbf{R}_{\text{total}}$ in Eq. 7 for reasoning quality, segmentation accuracy, and token efficiency. By evaluating rewards at the mini-batch group level, GRPO avoids the use of a reward model and stabilizes training. More details are provided in Sec. A of the Appendix.

### 3.4. Theoretical Analysis

In this section, we briefly analyze why this two-stage rollout improves RL-based reasoning segmentation from optimization and information-theoretic perspectives. The length-based reward is excluded to isolate the structural contribution. More details are provided in Sec. B of the Appendix.

**Optimization Analysis.** For clarity, we omit the format reward and the non-repetition reward in Eq. 6. Under standard reinforcement learning with answer-level supervision, the optimization objective is given by

$$\mathcal{L}_{base}(\theta) = \mathbb{E}_{\mathcal{S} \sim \pi_\theta}[\mathbf{R}_{acc}(\mathcal{A}, \mathcal{A}^*)], \qquad (9)$$

where $\mathcal{S} = (\mathcal{R}, \mathcal{A})$ is the response with reasoning chains $\mathcal{R}$, and $\pi_\theta$ is the policy MLLM. Since $\mathcal{R}$ is not directly supervised, the model is prone to unconstrained exploration in the reasoning space, often resulting in excessively long reasoning chains. From an optimization perspective, this behavior increases the stochasticity of sampled trajectories, leading to high-variance gradients.

To alleviate this issue, we decompose the reward into two complementary components: a description reward and an answer reward. The resulting optimization objective is

$$\mathcal{L}(\theta) = \mathbb{E}_{\mathcal{S} \sim \pi_\theta}\big[\mathbf{R}_{\text{desc}}(\mathcal{I}, \mathcal{D}) + \mathbf{R}_{acc}(A, A^*)\big], \qquad (10)$$

where $\mathbf{R}_{\text{desc}}$ provides intermediate supervision over descriptions $\mathcal{D}$, which are correlated with correct reasoning outcomes. By acting as an intermediate anchoring signal, $\mathbf{R}_{\text{desc}}$ improves credit assignment across the reasoning trajectory, thereby enabling more stable and efficient policy learning.

**Information-Theoretic Analysis.** Mutual information $\mathbf{I}(U; V)$ measures how much knowing $V$ reduces uncertainty about $U$ (Shannon, 1948). In a standard single-stage rollout, the dependency between the reasoning chain $\mathcal{R}$ and the final answer $\mathcal{A}$, conditioned on the image $\mathcal{I}$ and the query $\mathcal{Q}$, is quantified by conditional mutual information:

$$I(\mathcal{R}; \mathcal{A} \mid \mathcal{I}, \mathcal{Q}) = H(\mathcal{A} \mid \mathcal{I}, \mathcal{Q}) - H(\mathcal{A} \mid \mathcal{I}, \mathcal{Q}, \mathcal{R}). \quad (11)$$

This term measures how much information the reasoning chain provides for the answer beyond the given input image and question.

With the introduction of an intermediate description $\mathcal{D}$ in the two-stage rollout, the dependency becomes:

$$I(\mathcal{R}; \mathcal{A} \mid \mathcal{I}, \mathcal{Q}, \mathcal{D}) = H(\mathcal{A} \mid \mathcal{I}, \mathcal{Q}, \mathcal{D}) - H(\mathcal{A} \mid \mathcal{I}, \mathcal{Q}, \mathcal{D}, \mathcal{R}). \quad (12)$$

Here, $\mathcal{D}$ serves as a compact information bottleneck that preserves necessary information. Assuming that $\mathcal{D}$ is a sufficient statistic extracted from $\mathcal{R}$ with respect to $\mathcal{A}$, the conditional mutual information is reduced:

$$H(\mathcal{A} \mid \mathcal{I}, \mathcal{Q}, \mathcal{D}) \leq H(\mathcal{A} \mid \mathcal{I}, \mathcal{Q}), \qquad (13)$$

making the answer entropy of two-stage rollout reduced, as also supported by the experimental results in Fig. 4(a).

## 4. Experiments

### 4.1. Experimental Settings

**Dataset.** We first train the model using the 7K-sample setting of VisionReasoner (Liu et al., 2025b) for fair comparison, which is constructed from the LVIS (Gupta et al., 2019), RefCOCOg (Yu et al., 2016), gRefCOCO (Liu et al., 2023a), and LISA++ (Yang et al., 2023) datasets. We then follow the LISA (Lai et al., 2024) protocol and perform fine-tuning on the ReasonSeg train split, which contains only 239 samples with complex textual queries. For evaluation, we use the validation and test sets of ReasonSeg to evaluate performance in complex reasoning scenarios, and additionally report results on RefCOCO, RefCOCO+, and RefCOCOg to assess performance in referring segmentation.

**Evaluation Metrics.** Following prior work on reasoning segmentation (Lai et al., 2024), we adopt two evaluation metrics: gIoU and cIoU. gIoU is computed as the average per-image Intersection-over-Union (IoU), while cIoU computes the ratio of cumulative intersection to cumulative union across the dataset. Since cIoU is biased toward large-area objects and exhibits high variance, gIoU is used as the primary metric (Lai et al., 2024). In addition, we report the average number of reasoning tokens to measure efficiency.

**Experimental Details.** Following common practice (Liu et al., 2025b), we adopt Qwen2.5VL-7B (Bai et al., 2025) as the reasoning model and SAM2-Large (Ravi et al.) as the segmentation model. Reinforcement learning is performed using the GRPO algorithm (Shao et al., 2024). Training is conducted with a total batch size of 16 with 8-sample rollout per training step. The initial learning rate is set to 1e-6, and

*Table 1.* **Performance comparison on the ReasonSeg benchmark.** We additionally report the number of reasoning tokens to measure reasoning efficiency. * marks models trained on the train split of ReasonSeg. Bold and underlined values denote the best and second-best results, respectively. Our method shows notable superiority in both zero-shot and few-shot settings.

| Method | Language Model | ReasonSeg Val | | | ReasonSeg Test | | |
|---|---|---|---|---|---|---|---|
| | | Tokens ↓ | gIoU ↑ | cIoU ↑ | Tokens ↓ | gIoU ↑ | cIoU ↑ |
| OVSeg | CLIP ViT-L | – | 28.5 | 18.6 | – | 26.1 | 20.8 |
| ReLA | BERT | – | 22.4 | 19.9 | – | 21.3 | 22.0 |
| LISA | LLaVA1.5-7B | – | 53.6 | 52.3 | – | 48.7 | 48.8 |
| LISA* | LLaVA1.5-7B | – | 61.3 | 62.9 | – | 55.6 | 56.9 |
| CoReS | LLaVA-7B | – | 54.8 | – | – | 48.7 | – |
| CoReS* | LLaVA-7B | – | 59.4 | – | – | 52.4 | – |
| Seg-Zero | Qwen2.5VL-7B | 90.7 | 61.6 | 52.5 | 90.6 | 58.2 | 52.3 |
| SAM-R1 | Qwen2.5VL-7B | – | 64.0 | 55.8 | – | 60.2 | 54.3 |
| PixelThink | Qwen2.5VL-7B | 46.9 | 63.8 | 62.6 | 47.6 | 60.1 | 55.7 |
| VisionReasoner | Qwen2.5VL-7B | 80.8 | 66.3 | 59.8 | 84.8 | 63.6 | 58.2 |
| VisionReasoner* | Qwen2.5VL-7B | 85.3 | 65.4 | 60.3 | 81.4 | 62.3 | 54.6 |
| DR$^2$Seg (Ours) | Qwen2.5VL-7B | 46.2 | 67.5 | 60.0 | 55.4 | 64.8 | 62.8 |
| DR$^2$Seg* (Ours) | Qwen2.5VL-7B | **26.9** | **68.5** | **65.8** | **27.2** | **66.1** | **63.6** |

the weight decay is 0.01. The predefined length anchor $N_0$ is set to 45, and the length penalty parameter $\gamma$ is set to 0.05. Following (Liu et al., 2025b), we train the model for one epoch on VisionReasoner-7K in the zero-shot setting. We also train the model for five epochs on the ReasonSeg train set in the few-shot setting as (Lai et al., 2024). For all ablation studies, we train the model on the ReasonSeg train set and evaluate it on the ReasonSeg validation set. Notably, only the first-stage pass is used during evaluation to ensure computational efficiency and a fair comparison. The two-stage rollout is only employed during training to learn the referring description.

### 4.2. Main Results

**Comparison Methods.** As shown in Tab. 1 and Tab. 2, the comparison methods are organized by rows: the first row lists non-MLLM methods (OVSeg (Liang et al., 2023), LAVT (Yang et al., 2022), ReLA (Liu et al., 2023a)); the second row includes SFT-based methods (LISA (Lai et al., 2024), CORES (Bao et al., 2024), PixelLM (Ren et al., 2024), Perception-GPT (Pi et al., 2024)); the third row presents RL-based methods (Seg-Zero (Liu et al., 2025a), SAM-R1 (Huang et al., 2025), PixelThink (Wang et al., 2025)); and the last row reports the recent state-of-the-art VisionReasoner (Liu et al., 2025b), which also serves as the baseline for our method, together with our DR$^2$Seg.

**Reasoning Segmentation Results.** As shown in Tab. 1, we report the results on the ReasonSeg benchmark. In the zero-shot setting, DR$^2$Seg does not incorporate the length-based reward $\mathbf{R}_{\text{len}}$, since the training data predominantly consist of short referring expressions, for which Principle 2 cannot be reliably satisfied. Nevertheless, incorporating only the $\mathbf{R}_{\text{desc}}$

*Table 2.* **Performance comparison on referring expression segmentation.** * marks models trained on the train split of ReasonSeg, and their performance is reported to evaluate generalization.

| Method | refCOCO | refCOCO+ | refCOCOg |
|---|---|---|---|
| | testA ↑ | testA ↑ | test ↑ |
| LAVT | 75.8 | 68.4 | 62.1 |
| ReLA | 76.5 | 71.0 | 66.0 |
| LISA-7B | 76.5 | 67.4 | 68.5 |
| PixelLM-7B | 76.5 | 71.7 | 70.5 |
| Perception-GPT-7B | 78.6 | 73.9 | 71.7 |
| Seg-Zero-7B | **80.3** | **76.2** | 72.6 |
| PixelThink-7B | 79.3 | 74.8 | **73.9** |
| VisionReasoner-7B | 78.9 | 74.9 | 71.3 |
| VisionReasoner*-7B | 78.8 | 75.1 | 71.5 |
| DR$^2$Seg-7B (Ours) | 78.7 | 75.4 | 72.2 |
| DR$^2$Seg*-7B (Ours) | 79.3 | 75.4 | 73.4 |

reward nearly halves the number of inference tokens and significantly outperforms VisionReasoner under the same zero-shot setting, achieving gIoU improvements of 1.2% on the validation set and 1.2% on the test set. Furthermore, we perform few-shot fine-tuning using the ReasonSeg train data with only 239 samples. Directly training VisionReasoner on the ReasonSeg train set leads to accuracy degradation. In contrast, our DR$^2$Seg* consistently improves segmentation accuracy, achieving gIoU scores of 68.5% and 66.1% on the validation and test sets, achieving a new state-of-the-art. These results indicate that our method can effectively learn to decouple complex reasoning segmentation with limited data by self-rewarding the reasoning process. Consequently, DR$^2$Seg* outperforms VisionReasoner* by 3.1% on the val-

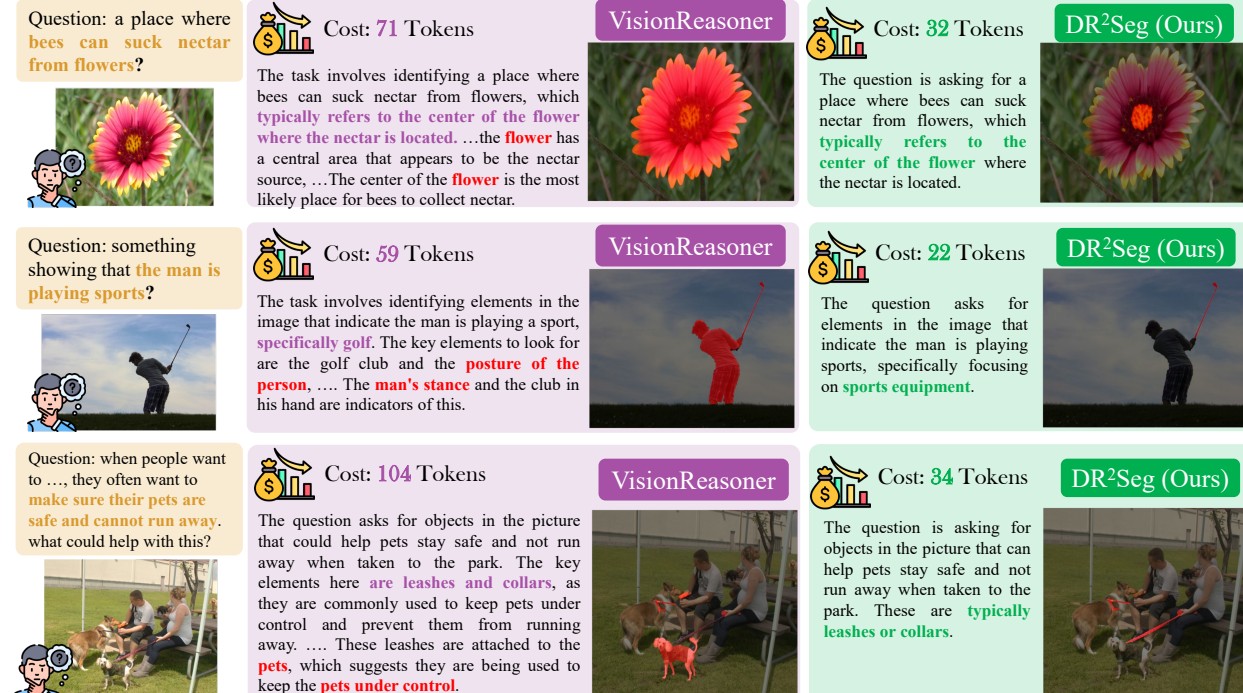

*Figure 3.* **Qualitative comparisons between VisionReasoner and our DR²Seg.** The representative samples are selected from simple single-object to complex multi-object scenarios.

*Table 3.* **Ablation studies of DR²Seg.** $R_{desc}$ and $R_{len}$ denote the description self-reward and length-based self-reward, respectively.

| $R_{desc}$ | $R_{len}$ | Tokens ↓ | gIoU ↑ | cIoU ↑ |
|---|---|---|---|---|
| | | 81.5 | 64.9 | 58.8 |
| ✓ | | 56.1 | 67.6 | 63.1 |
| ✓ | ✓ | 26.9 | 68.5 | 65.8 |

idation set and 3.8% on the test set in gIoU, while reducing the number of tokens by 3×.

**Referring Segmentation Results.** We further report results on the referring expression segmentation (RES) task to evaluate the generalization ability of reasoning models in relatively simple scenarios, as shown in Tab. 2. Clearly, our DR²Seg method also demonstrates consistently strong performance, indicating that the self-rewarding framework does not impair the model's ability to handle simple scenarios. Furthermore, after fine-tuning on the ReasonSeg train set, DR²Seg* even improves upon DR²Seg, which is trained only on RES train datasets, further validating the generalization of our method. Overall, DR²Seg* outperforms the baseline VisionReasoner*, achieving a 1.9% improvement on the refCOCOg dataset.

**Qualitative Results.** Fig. 3 presents a qualitative comparison between VisionReasoner and DR²Seg. VisionRea-

soner exhibits an *overthinking* issue: although it correctly identifies the target object, excessive reasoning leads to attention confusion in subsequent perception (e.g., the red-highlighted text misguides the MLLM toward erroneous regions). In contrast, DR²Seg reduces reasoning tokens while maintaining focus on the target, achieving a better balance between efficiency and accuracy. More qualitative results are provided in Sec. D of the Appendix.

### 4.3. Diagnostic Experiments

**Ablation on DR²Seg scheme.** We ablate the two core self-rewards of DR²Seg in Tab. 3: the description self-reward $R_{desc}$ and the length-based self-reward $R_{len}$, both derived from our two-stage rollout strategy. By decoupling reasoning segmentation into multimodal reasoning and referring segmentation, the MLLM attains clearer task objectives and stronger focus on the target object. $R_{desc}$ leads to a notable improvement in accuracy, whereas the incorporation of $R_{len}$ not only reduces the count of reasoning tokens but also yields an extra accuracy gain, thus validating the effectiveness of our two-stage design principle.

**Effect of Two-stage Rollout.** To further analyze the effectiveness of the two-stage rollout, we examine the evolution of answer entropy, reasoning token count, and accuracy during training, as shown in Fig. 4. For answer entropy computation, we measure only the entropy of segmentation answer tokens, excluding other reasoning tokens, to cap-

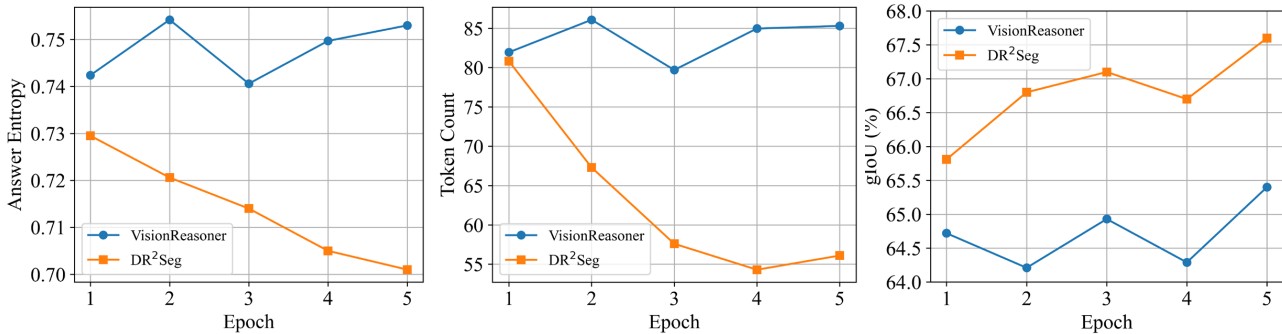

*Figure 4.* **Effect of the Two-Stage Rollout Strategy.** We analyze the evolution of answer entropy, thinking token count, and accuracy during training, where answer entropy reflects the model's output uncertainty.

*Table 4.* **Ablation on length anchor of length-based reward.**

| $N_0$ | Tokens ↓ | gIoU ↑ | cIoU ↑ |
|-------|----------|--------|--------|
| 55 | 44.4 | 66.8 | 55.3 |
| 45 | 26.9 | 68.5 | 65.8 |
| 35 | 20.2 | 66.6 | 58.0 |
| 25 | 70.7 | 66.3 | 57.7 |

*Table 5.* **Ablation on length penalty of length-based reward.**

| $\gamma$ | Tokens ↓ | gIoU ↑ | cIoU ↑ |
|----------|----------|--------|--------|
| 0.01 | 44.4 | 66.8 | 55.3 |
| 0.05 | 26.9 | 68.5 | 65.8 |
| 0.1 | 26.8 | 68.9 | 64.9 |
| 0.2 | 26.7 | 68.0 | 60.5 |

*Table 6.* **Performance evaluation on smaller 3B MLLMs.**

| Method | Tokens ↓ | gIoU ↑ | cIoU ↑ |
|--------|----------|--------|--------|
| VisionReasoner-3B | 49.7 | 61.7 | 54.2 |
| DR$^2$Seg-3B | 23.9 | 65.5 | 56.8 |

*Table 7.* **Performance evaluation on the more recent segmentation Model SAM3 ([Carion et al., 2025](#)).**

| Method | Seg. Model | Tokens ↓ | gIoU ↑ | cIoU ↑ |
|--------|-----------|----------|--------|--------|
| VisionReasoner | SAM3 | 64.8 | 65.8 | 61.5 |
| DR$^2$Seg $-$ $\mathbf{R}_{len}$ | SAM3 | 63.7 | 68.7 | 64.8 |
| DR$^2$Seg | SAM3 | 31.2 | 69.4 | 66.4 |

ture the MLLM's uncertainty in localization. This entropy is computed only over the final answer tokens, excluding reasoning tokens, so it specifically reflects the model's confidence in localization rather than variability in the reasoning trace itself. Notably, the length-based reward is not applied, allowing us to isolate the effect of the two-stage rollout structure. As training progresses, both answer entropy and reasoning token count consistently decrease, while accuracy steadily improves. These trends indicate that decoupling reasoning segmentation reduces overthinking, enables more confident target localization, and ultimately improves both accuracy and efficiency.

**Ablation on Length Anchor.** We analyze the effect of different values of length anchor $N_o$. As shown in Tab. 4, reducing $N_o$ generally leads to fewer reasoning tokens. When $N_o$ decreases from 55 to 45, reasoning becomes shorter while accuracy improves, suggesting that the original reasoning contains redundancy. Reducing $N_o$ from 45 to 35 further compresses reasoning, but slightly hurts accuracy due to over-compression. However, when $N_o$ is set too small (e.g., 25), the number of reasoning tokens increases

sharply. This indicates that when $N_o$ is overly small, the model fails to infer the target object, leading to a conflict between the accuracy and length rewards. Specifically, under Eq. 8, when the accuracy reward drops to zero, the length reward becomes ineffective and no longer constrains the reasoning length. To obtain a suitable $N_o$, we first measure the average reasoning token without the length-based self-reward (e.g., 46 tokens in DR²Seg seeing Tab. 1), and then choose candidate values around that average. In practice, a suitable can be identified within only 3-4 trials.

**Ablation on Length Penalty.** We ablate different values of the length penalty $\gamma$, as shown in Tab. 5. When $\gamma$ is set to a small value (e.g., 0.01), the penalty for exceeding the length constraint is insufficient, resulting in long sequences and reduced accuracy. Once $\gamma$ is properly chosen, both token length and accuracy remain stable, indicating that $\gamma$ is not a sensitive hyperparameter. The model can effectively learn to respect the length constraint during training.

**Ablation on Smaller Model.** We also report results on smaller 3B-parameter MLLMs, as shown in Tab. 6. DR$^2$Seg achieves $\sim 2\times$ reduction in thinking tokens and a notable 3.8% gIoU improvement, demonstrating its effectiveness across MLLMs of different scales.

*Table 8.* **Performance evaluation on single-pass and second-pass of DR$^2$Seg.**

| Method | gIoU ↑ | cIoU ↑ |
|---|---|---|
| $\mathcal{D}$−only (Second-Pass) | 65.9 | 62.6 |
| DR$^2$Seg (Single-Pass) | 66.1 | 63.6 |

**Ablation on SAM3.** We further evaluate our method on the recent segmentation model, SAM3 (Carion et al., 2025), which supports concept segmentation with brief phrases. Unlike SAM2 (Ravi et al.), which directly supervises box and point predictions, we employ an external SAM3 API as a reward generator during training. Phrase descriptions produced by the MLLM are fed into SAM3 to obtain segmentation results, from which a IoU-based reward is computed (see Sec. C.3 of the Appendix). Owing to the two-stage rollout strategy, our method naturally generates brief descriptions that align well with SAM3. As shown in Tab. 7, DR$^2$Seg achieves significant improvements in both accuracy and efficiency, demonstrating its versatility.

**Ablation on Single-Pass and Second-Pass.** To address whether the second-pass self-verification actually transfers to the first-pass outputs, we further evaluated the performance of second-pass using only description $\mathcal{D}$ as input on the ReasonSeg-test dataset. As shown in Tab. 8, the results were still promising even without the question query. This demonstrates that the learned description is able to accurately refer to the target.

## 5. Conclusions

In this paper, we propose a self-reward framework featuring a two-stage rollout strategy that effectively decouples reasoning segmentation into multimodal reasoning and referring segmentation. Building on this design, we introduce self-rewards that self-supervise reasoning, preventing MLLMs from being misled by redundant reasoning and thereby improving both efficiency and accuracy. Extensive experiments across MLLMs of different scales, diverse segmentation models, and both complex reasoning and simple referring scenarios demonstrate the effectiveness of our method. This work provides new insights into efficient and accurate reasoning perception with MLLMs.

## Impact Statement

Regarding the datasets, all datasets used in this paper are publicly available and have undergone appropriate ethical review and approval. With respect to the proposed algorithm, DR$^2$Seg enables accurate and robust reasoning segmentation with high efficiency. By effectively decoupling reasoning segmentation into multimodal reasoning and perception, it provides a powerful framework for advancing the synergy between reasoning and perception, with strong potential to support cognitive and perceptual capabilities in domains such as medical applications and embodied intelligence.

## Acknowledgments

This work was supported by a grant from National Science Fund for Distinguished Young Scholars of China (No. 62525213), and the Innovation Research Foundation of National University of Defense Technology (No. ZK25-11).

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

# A. Group-Relative Policy Optimization

In this section, we introduce the details of Group-Relative Policy Optimization (GRPO) (Shao et al., 2024), a policy gradient method designed to improve stability and scalability in LLM optimization by leveraging relative comparisons among multiple samples generated under the same context.

## A.1. Problem Setup

Given an input context (or state) $x \in \mathcal{X}$, a parameterized policy $\pi_\theta(y \mid x)$ generates an output $y \in \mathcal{Y}$. A reward function $r(x, y)$ evaluates the quality of the output. The policy optimization objective is defined as:

$$\max_\theta \ \mathbb{E}_{x \sim \mathcal{D}, \, y \sim \pi_\theta(\cdot \mid x)} \left[ r(x, y) \right]. \tag{14}$$

## A.2. Group Sampling

For each context $x$, we sample a group of $K$ outputs from the current policy:

$$\{y_1, y_2, \ldots, y_K\} \sim \pi_\theta(\cdot \mid x). \tag{15}$$

Each sampled output $y_i$ is assigned a reward:

$$r_i = r(x, y_i). \tag{16}$$

## A.3. Group-Relative Advantage

We define the group mean reward as:

$$\bar{r} = \frac{1}{K} \sum_{j=1}^{K} r_j. \tag{17}$$

The group-relative advantage is computed via mean-centering:

$$A_i = r_i - \bar{r}. \tag{18}$$

Optionally, variance normalization can be applied:

$$A_i = \frac{r_i - \bar{r}}{\sqrt{\frac{1}{K} \sum_{j=1}^{K} (r_j - \bar{r})^2 + \epsilon}}. \tag{19}$$

This construction guarantees a zero-sum property within each group:

$$\sum_{i=1}^{K} A_i = 0. \tag{20}$$

## A.4. GRPO Objective

The Group-Relative Policy Optimization objective is defined as:

$$\mathcal{L}_{\text{GRPO}}(\theta) = \mathbb{E}_{x \sim \mathcal{D}} \left[ \frac{1}{K} \sum_{i=1}^{K} A_i \log \pi_\theta(y_i \mid x) \right]. \tag{21}$$

To constrain policy updates, GRPO introduces a KL regularization term with respect to a reference policy $\pi_{\text{ref}}$:

$$\mathcal{L}(\theta) = \mathcal{L}_{\text{GRPO}}(\theta) - \beta \, \mathbb{E}_x \left[ \text{KL}(\pi_\theta(\cdot \mid x) \, \| \, \pi_{\text{ref}}(\cdot \mid x)) \right], \tag{22}$$

where $\beta$ controls the strength of the regularization.

GRPO differs from conventional policy optimization methods in that it does not rely on an explicit value function. Instead, the group mean reward acts as a data-driven baseline, yielding low-variance gradient estimates without additional learned components. Moreover, by focusing on relative performance within a group, GRPO naturally aligns with ranking-based supervision and preference learning, making it particularly suitable for large-scale models where multiple candidate outputs per input are readily available.

## B. More Theoretical Analysis

### B.1. Detailed Optimization Analysis

Let $\pi_\theta$ denote the policy (the MLLM) parameterized by $\theta$ that samples a response (trajectory) $\mathcal{S} = (\mathcal{R}, \mathcal{A})$, where $\mathcal{R}$ is the reasoning chain and $\mathcal{A}$ the final answer. Let $\mathbf{R}_{\mathrm{acc}}(\mathcal{A}, \mathcal{A}^*)$ denote the answer-level accuracy reward and let $\mathbf{R}_{\mathrm{desc}}(\mathcal{I}, \mathcal{D})$ denote an intermediate description reward defined on the description $\mathcal{D}$ produced in stage 1. The baseline objective (answer-only supervision) is

$$\mathcal{L}_{\mathrm{base}}(\theta) = \mathbb{E}_{\mathcal{S} \sim \pi_\theta}\big[\mathbf{R}_{\mathrm{acc}}(\mathcal{A}, \mathcal{A}^*)\big]. \tag{23}$$

With the two-stage reward decomposition we optimize

$$\mathcal{L}(\theta) = \mathbb{E}_{\mathcal{S} \sim \pi_\theta}\big[\mathbf{R}_{\mathrm{desc}}(\mathcal{I}, \mathcal{D}) + \mathbf{R}_{\mathrm{acc}}(\mathcal{A}, \mathcal{A}^*)\big]. \tag{24}$$

**Policy-gradient form and variance.** Using the score-function identity, the gradient of either objective can be written as an expectation over trajectories:

$$\nabla_\theta \mathcal{L}(\theta) = \mathbb{E}_{\mathcal{S} \sim \pi_\theta}\big[G(\mathcal{S})\,\nabla_\theta \log \pi_\theta(\mathcal{S})\big], \tag{25}$$

where $G(\mathcal{S})$ is the scalar return used by the objective. Under the base objective $G_{\mathrm{base}}(\mathcal{S}) = \mathbf{R}_{\mathrm{acc}}(\mathcal{A}, \mathcal{A}^*)$, whereas under the decomposed objective

$$G_{\mathrm{decomp}}(\mathcal{S}) = \mathbf{R}_{\mathrm{desc}}(\mathcal{I}, \mathcal{D}) + \mathbf{R}_{\mathrm{acc}}(\mathcal{A}, \mathcal{A}^*). \tag{26}$$

For Monte Carlo gradient estimation, the variance of the estimator is governed by the variance of the scalar multiplier $G(\mathcal{S})$ (and by the variance of the score function). A useful approximation (common in practice) is that

$$\mathrm{Var}\big[G(\mathcal{S})\,\nabla_\theta \log \pi_\theta(\mathcal{S})\big] \approx \mathbb{E}\big[\|\nabla_\theta \log \pi_\theta(\mathcal{S})\|^2\big] \cdot \mathrm{Var}\big[G(\mathcal{S})\big]. \tag{27}$$

Consequently, reducing $\mathrm{Var}[G(\mathcal{S})]$ reduces the gradient variance and typically improves sample efficiency and stability.

Using Eq. (26) and the variance decomposition,

$$\mathrm{Var}\big[G_{\mathrm{decomp}}\big] = \mathrm{Var}\big[\mathbf{R}_{\mathrm{acc}}\big] + \mathrm{Var}\big[\mathbf{R}_{\mathrm{desc}}\big] + 2\,\mathrm{Cov}\big(\mathbf{R}_{\mathrm{acc}}, \mathbf{R}_{\mathrm{desc}}\big). \tag{28}$$

If the description reward $\mathbf{R}_{\mathrm{desc}}$ captures predictable, answer-relevant signal (i.e., it is positively correlated with the "goodness" of trajectories), and in particular if it explains a substantial portion of the variability in $\mathbf{R}_{\mathrm{acc}}$, then adding $\mathbf{R}_{\mathrm{desc}}$ can reduce the overall variance of the return used in the gradient estimator. Intuitively, $\mathbf{R}_{\mathrm{desc}}$ acts like a control variate: it explains part of the final reward so less stochasticity remains for the Monte Carlo estimator to resolve. Whether $\mathrm{Var}[G_{\mathrm{decomp}}]$ is smaller than $\mathrm{Var}[G_{\mathrm{base}}]$ depends on the magnitudes and covariance in Eq. (28). This motivates empirical measurement of the covariance term.

**Connection to control variates and potential-based shaping.** Two standard variance-reduction mechanisms are relevant.

(i) *Control variate / baseline.* Subtracting a baseline $b$ from the return does not change the expected gradient but reduces variance:

$$\nabla_\theta \mathcal{L}(\theta) = \mathbb{E}\big[(G(\mathcal{S}) - b(\mathcal{S}))\,\nabla_\theta \log \pi_\theta(\mathcal{S})\big]. \tag{29}$$

If $\mathbf{R}_{\mathrm{desc}}$ is strongly correlated with $\mathbf{R}_{\mathrm{acc}}$, it can play a similar role to a data-dependent baseline by explaining predictable parts of the return.

(ii) *Potential-based reward shaping.* Let $\Phi(s)$ be a scalar potential on states. The shaping reward

$$F(s_t, s_{t+1}) = \gamma \Phi(s_{t+1}) - \Phi(s_t) \tag{30}$$

is known to preserve optimal policies when added to any MDP reward (i.e., it is policy invariant). If $\mathbf{R}_{\text{desc}}$ can be expressed (or approximated) as a sum of potential differences across intermediate states, then it will not change the optimal policy while changing the learning dynamics to be more informative at intermediate steps. In practice, an intermediate description reward that provides informative intermediate feedback can be viewed as a form of reward shaping that improves credit assignment and reduces exploration noise, while leaving the optimal solution intact under the potential-based condition.

### B.2. Detailed Information-Theoretic Analysis

We give a principled information-theoretic account of why the proposed two-stage rollout (stage-1: produce description $\mathcal{D}$; stage-2: reason on $(\mathcal{I}, \mathcal{D})$) can yield more stable and more concise reasoning than a single-stage rollout.

Notation Image: $\mathcal{I}$. Original query: $\mathcal{Q}$. Stage-1 reasoning chain and intermediate output: $\mathcal{R}^1$ and $\mathcal{D} = g(\mathcal{I}, \mathcal{Q}, \mathcal{R}^1)$. Stage-2 reasoning chain and final answer: $\mathcal{R}^2$ and $\mathcal{A}^2$. Stage-2 consumes $(\mathcal{I}, \mathcal{D})$.

1) Mutual-information formulation for stage-2

$$I(\mathcal{R}^2; \mathcal{A}^2 \mid \mathcal{I}, \mathcal{D}) = H(\mathcal{A}^2 \mid \mathcal{I}, \mathcal{D}) - H(\mathcal{A}^2 \mid \mathcal{I}, \mathcal{D}, \mathcal{R}^2). \tag{31}$$

2) Data-processing inequality and the role of $\mathcal{D}$ Assume the pipeline satisfies the conditional Markov chain

$$\mathcal{R}^1 \longrightarrow \mathcal{D} \longrightarrow (\mathcal{R}^2, \mathcal{A}^2) \quad \text{given } \mathcal{I}, \tag{32}$$

i.e., stage-2 depends on stage-1 only via $\mathcal{D}$. By the data-processing inequality (DPI),

$$I(\mathcal{R}^1; \mathcal{A}^2 \mid \mathcal{I}) \geq I(\mathcal{D}; \mathcal{A}^2 \mid \mathcal{I}). \tag{33}$$

If $\mathcal{D}$ is deterministic from $(\mathcal{I}, \mathcal{Q}, \mathcal{R}^1)$ and the Markov property holds, then

$$I(\mathcal{R}^1; \mathcal{A}^2 \mid \mathcal{I}) = I(\mathcal{D}; \mathcal{A}^2 \mid \mathcal{I}). \tag{34}$$

DPI therefore formalizes that compressing $\mathcal{R}^1$ into $\mathcal{D}$ cannot increase the information about the final answer.

3) Information-bottleneck (IB) justification for $\mathcal{D}$ To make the bottleneck claim nontrivial, we adopt an IB perspective and posit that $\mathcal{D}$ is constructed to trade off compression of inputs and preservation of predictive information:

$$\min_{p(\mathcal{D}|\mathcal{I}, \mathcal{Q}, \mathcal{R}^1)} I(\mathcal{D}; \mathcal{I}, \mathcal{Q}) \quad \text{s.t.} \quad I(\mathcal{D}; \mathcal{A}^2) \geq \alpha, \tag{35}$$

for some threshold $\alpha$. Under this objective, $\mathcal{D}$ discards variability in $(\mathcal{I}, \mathcal{Q}, \mathcal{R}^1)$ that is irrelevant to predicting $\mathcal{A}^2$. This is stronger than the trivial inequality

$$H(\mathcal{A}^2 \mid \mathcal{I}, \mathcal{D}) \leq H(\mathcal{A}^2 \mid \mathcal{I}, \mathcal{Q}), \tag{36}$$

because IB enforces $\mathcal{D}$ to be a compact, task-oriented summary rather than any arbitrary variable.

From IB and DPI we obtain the nontrivial relation

$$I(\mathcal{D}; \mathcal{A}^2) \leq I(\mathcal{R}^1; \mathcal{A}^1), \tag{37}$$

and $\mathcal{D}$ is optimized to minimize irrelevant input information while retaining predictive power.

4) From reduced uncertainty to conciseness. To connect entropy reduction with token length, use standard coding bounds. For vocabulary size $|\mathcal{V}|$, the expected token length satisfies, up to constants,

$$\mathbb{E}[\text{len}(\mathcal{R})] \gtrsim \frac{H(\mathcal{R} \mid \cdot)}{\log_2 |\mathcal{V}|}. \tag{38}$$

Hence, if the IB-trained $\mathcal{D}$ reduces the conditional entropy of the second-stage chain,

$$H(\mathcal{R}^2 \mid \mathcal{I}, \mathcal{D}) < H(\mathcal{R}^1 \mid \mathcal{I}, \mathcal{Q}), \tag{39}$$

then, under near-optimal encoding/decoding, the expected token length of $\mathcal{R}^2$ can be smaller than that of $\mathcal{R}^1$. Note this requires (i) $\mathcal{D}$ actively filters irrelevant variability (IB) and (ii) generation efficiency is sufficient to reflect entropy reductions in token lengths.

5) Assume: (A1) $\mathcal{D} = g(\mathcal{I}, \mathcal{Q}, \mathcal{R}^1)$ (possibly stochastic); (A2) conditional Markov chain $\mathcal{R}^1 - \mathcal{D} - (\mathcal{R}^2, \mathcal{A}^2)$ given $\mathcal{I}$; (A3) $\mathcal{D}$ is obtained under an IB-type objective; (A4) generation is reasonably efficient relative to entropy bounds.

Then DPI + IB imply $\mathcal{D}$ preserves task-relevant information while discarding irrelevant variability, yielding reduced $H(\mathcal{R}^2 \mid \mathcal{I}, \mathcal{D})$ and, consequently, potentially shorter expected $\mathcal{R}^2$ (i.e., more concise reasoning) and more stable stage-2 predictions.

When $\mathcal{D}$ is explicitly optimized as a compact, task-relevant summary (IB principle) and the pipeline satisfies the stated Markov assumptions, the two-stage rollout admits a nontrivial information-theoretic explanation: $\mathcal{D}$ filters irrelevant variability, reducing stage-2 uncertainty and enabling more concise and stable reasoning in practice.

## C. Additional Implementation Details.

In this section, we provide additional implementation details for the reward details, the prompt designs, and the SAM3 implementation in the main paper.

### C.1. Reward Details

The base reward $\mathbf{R}_{\text{base}}$ from VisionReasoner (Liu et al., 2025b) is defined as:

**Format Reward.** This reward enforces a structured output: the model must place its chain-of-thought (the reasoning chains) between the markers. `<think> ... </think>` and `<answer> ... </answer>`. We represent answers using 2D bounding boxes and 2D points. Concretely, use the collections of bounding boxes $(\mathbf{B}_i)_{i=1}^N$ and points $(\mathbf{P}_i)_{i=1}^N$. The model's textual output should follow the JSON-like format: ["bbox_2d": $[x_1, y_1, x_2, y_2]$, "point_2d": $[x_1, y_1]$, ...].

**Non-Repeat Reward.** To discourage repeated patterns, split the reasoning process into individual sentences and prioritize sentences that are unique or non-repetitive. The reward favors diversity in the sentence-level reasoning steps.

**Accuracy Reward.** Accuracy Reward includes Bboxes IoU Reward, Bboxes L1 Reward, and Points L1 Reward.

- Bboxes IoU Reward. Let $\{\mathbf{B}_i\}_{i=1}^N$ be the ground-truth bounding boxes and $\{\hat{\mathbf{B}}_j\}_{j=1}^K$ the predicted bounding boxes. Compute an optimal one-to-one matching $\mathcal{M}$ between ground-truth and predicted boxes (e.g., Hungarian matching that maximizes total IoU). For each matched pair $(i, j) \in \mathcal{M}$ whose Intersection-over-Union exceeds 0.5, add $\frac{1}{\max\{N,K\}}$ to the reward.

- Bboxes L1 Reward. Using the same one-to-one matching $\mathcal{M}$ between ground-truth and predicted boxes, compute the L1 distance between matched box coordinates. For each matched pair whose L1 distance is below 10 pixels, add $\frac{1}{\max\{N,K\}}$ to the reward.

- Points L1 Reward. Let $\{\mathbf{P}_i\}_{i=1}^N$ be the ground-truth points and $\{\hat{\mathbf{P}}_j\}_{j=1}^K$ the predicted points. Using the same one-to-one matching $\mathcal{M}$, compute the L1 distance between matched points. For each matched pair whose L1 distance is below 30 pixels, add $\frac{1}{\max\{N,K\}}$ to the reward.

### C.2. Prompt Designs

Tab. 9 and Tab. 10 present the prompt configurations of VisionReasoner and $\text{DR}^2\text{Seg}$, respectively. To rigorously validate the effectiveness of the proposed method, the prompt of $\text{DR}^2\text{Seg}$ is modified only by adding a description component, while all other parts are kept unchanged. During the two-stage rollout training of $\text{DR}^2\text{Seg}$, the content of the description is extracted to replace the original "Question" when generating the second-round response. This response is then used to compute self-rewards, enabling self-supervision that encourages the model to reason more efficiently and perform more accurate segmentation.

### C.3. Implementation Details on SAM3

In this paper, the training process leverages the SAM3 model via an external API, which takes Base64-encoded images, bounding box coordinates, and text prompts as input and returns Base64-encoded binary segmentation masks. To comply with SAM3's input specifications, we implement a coordinate conversion module within the API that transforms SAM2-style bounding boxes (represented by top-left and bottom-right coordinates $[x_1, y_1, x_2, y_2]$) into the normalized center-based format $[c_x, c_y, w, h]$ required by SAM3, while filtering out invalid boxes.

Please find "Question" with bboxs and points.

Compare the difference between object(s) and find the most closely matched object(s).

Output the thinking process in <think> </think>, the explicit referring description for object localization in <description> </description>, and final answer in <answer> </answer> tags.

Output the bbox(es) and point(s) inside the interested object(s) in JSON format.

i.e., <think>thinking process here </think>

<description>referring description here </description>

<answer>["bbox_2d": [10,100,200,210], "point_2d": [30,110], "bbox_2d": [225,296,706,786], "point_2d": [302,410]]</answer>

*Table 9.* Prompt template for VisionReasoner.

Please find "Question" with bboxs and points.

Compare the difference between object(s) and find the most closely matched object(s).

Output the thinking process in <think> </think>, the explicit referring description for object localization in <description> </description>, and final answer in <answer> </answer> tags.

Output the bbox(es) and point(s) inside the interested object(s) in JSON format.

i.e., <think>thinking process here </think>

<description>referring description here </description>

<answer>["bbox_2d": [10,100,200,210], "point_2d": [30,110], "bbox_2d": [225,296,706,786], "point_2d": [302,410]]</answer>

*Table 10.* Prompt template for DR$^2$Seg.

For mask generation, we exploit SAM3's strong text-understanding capability by designing a dual-prompt strategy that combines textual descriptions with bounding boxes, without relying on point prompts. Specifically, when the reasoning model fails to produce valid bounding boxes, the API generates segmentation masks solely based on the textual description extracted from the `<target_desc>` tag. When valid bounding boxes are available, the API iterates over each box independently, resetting the model's prompt state before each iteration to prevent cross-target interference. Each mask is generated using a combination of the box and the corresponding text prompt, and the final prediction is obtained by merging all individual masks through a logical OR operation.

To ensure the quality and consistency of the text prompts, we introduce explicit `<target_desc>` and `</target_desc>` tags during prompt engineering, enforcing the reasoning model to output concise and discriminative target phrases augmented with spatial or attribute cues (e.g., "the cup on the left side of the frame"). This design effectively bridges the semantic gap between free-form natural language reasoning and the segmentation model's prompt interpretation.

Regarding reward function design, we define a segmentation IoU reward by directly computing the Intersection over Union (IoU) between the merged predicted mask returned by the API and the ground-truth mask. This IoU value serves as the primary accuracy reward. Together with a format reward that verifies the structural integrity of outputs (e.g., correct use of `<target_desc>` tags) and a non-repetition reward, it forms the final composite reward used to guide the reasoning model toward generating outputs that are more amenable to accurate segmentation by SAM3.

## D. Additional Analysis

### D.1. Analysis of Training Efficiency

We compare the training efficiency of VisionReasoner, which adopts a standard single-stage rollout, with that of the proposed DR$^2$Seg, which employs an efficient two-stage rollout strategy. All experiments are conducted under identical conditions using 8 L40 GPUs. The models are trained on the ReasonSeg train dataset for 5 epochs with a batch size of 16, resulting in a total of 70 training iterations. VisionReasoner requires 3 hours and 15 minutes to complete training, whereas DR$^2$Seg

*Table 11.* **Additional performance comparison on the ReasonSeg benchmark with different scales of MLLM and vary segmentation models.** We report the number of reasoning tokens to measure reasoning efficiency. Bold values denote the best results.

| Method | Language Model | Segmentation Model | ReasonSeg Val | | | ReasonSeg Test | | |
|---|---|---|---|---|---|---|---|---|
| | | | Tokens ↓ | gIoU ↑ | cIoU ↑ | Tokens ↓ | gIoU ↑ | cIoU ↑ |
| VisionReasoner | Qwen2.5VL-3B | SAM2 | 49.7 | 61.7 | 54.2 | 49.3 | 58.0 | 51.2 |
| DR$^2$Seg (Ours) | Qwen2.5VL-3B | SAM2 | **23.9** | **65.5** | **56.8** | **33.3** | **60.2** | **55.0** |
| VisionReasoner | Qwen2.5VL-7B | SAM3 | 64.8 | 65.8 | 61.5 | 64.7 | 65.5 | 59.2 |
| DR$^2$Seg (Ours) | Qwen2.5VL-7B | SAM3 | **31.2** | **69.4** | **66.4** | **30.7** | **66.5** | **61.7** |

finishes in 3 hours and 10 minutes. Overall, the training efficiency of the two methods is comparable.

Although the two-stage rollout in DR$^2$Seg introduces an 2× increase in the number of rollouts, it performs inference using the model itself and thus does not require loading additional models. Furthermore, the proposed self-reward design significantly reduces the number of reasoning tokens, which further accelerates training. Consequently, the overall training efficiency of DR$^2$Seg is on par with that of the baseline VisionReasoner model employing a single-stage rollout strategy.

### D.2. Method Generalization Analysis

As shown in Tab. 11, we further evaluate the proposed DR$^2$Seg and the baseline VisionReasoner on ReasonSeg test using MLLMs of different model scales as well as different segmentation modules. The results are consistent with those reported in Tab. 6 and Tab. 7 in the main paper. On the ReasonSeg test set, the proposed DR$^2$Seg consistently achieves performance improvements across all configurations, demonstrating its strong generalization capability and versatility.

### D.3. Description Analysis

Fig. 5 illustrates the reasoning process, the generated descriptions, and the final segmentation results for different examples. Overall, the generated descriptions are typically expressed in the form of a single word or a short phrase, whose length depends on whether additional clarification is required to distinguish the target object.

For instance, in Fig. (a), the query asks about the girl's trainer, while three people appear in the image. Through its reasoning process, the MLLM produces the description "adult holding hand", which is both semantically precise and concise, enabling clear identification of the target. In contrast, for cases with little or no ambiguity, such as the "dragon boats" in Fig. (b), no additional modifiers are necessary. Fig. (c) demonstrates the model's ability to refer to a specific part of an object, such as the "stamen" of a flower. In Fig. (d), since the query itself is already sufficiently explicit, the generated description remains consistent with the original question after reasoning, without introducing incorrect inferences. This behavior also explains why our method performs well on simpler referring segmentation benchmarks even after training. In Fig. (e), the description includes the modifier "long", which helps the model segment the "fabric" as a whole rather than only a small local region.

In summary, these examples show that the generated descriptions are adaptive and target-oriented: when the target is unambiguous, a single word suffices, whereas additional modifiers are automatically introduced when finer discrimination is needed. This qualitative analysis demonstrates the strong adaptability and generalization capability of our method.

### D.4. More Visualization Analysis

To validate the performance of our method across diverse scenarios, we further select representative samples for qualitative visualization. As shown in Fig. 6(a), our method is able to segment highly irregular objects, which constitute particularly challenging segmentation cases. Fig. 6(b) demonstrates the capability to segment local regions of multiple objects. In Fig. 6(c), our method successfully segments objects that are blurred or camouflaged. Fig. 6(d) illustrates effective segmentation in multi-object scenes. Fig. 6(f) shows that our method can correctly segment objects appearing in reflections or mirrors, while Fig. 6(g) highlights its ability to detect and segment extremely small targets.

These qualitative results collectively demonstrate the strong reasoning segmentation capability of our method across a wide range of challenging scenarios.

**D.5. Optimization Analysis with Length-Based Self-Reward**

Let $\pi_\theta$ denote the policy parameterized by $\theta$, which samples a trajectory $\mathcal{S}$ containing the first-pass reasoning chain $\mathcal{R}^1$, the intermediate description $\mathcal{D}$, the second-pass reasoning chain $\mathcal{R}^2$, and the final answer $\mathcal{A}$. With the proposed reward design, the total reward is

$$\mathbf{R}_{\text{total}}(\mathcal{S}) = \big(\mathbf{R}_{\text{base}}(\mathcal{S}) + \mathbf{R}_{\text{desc}}(\mathcal{S})\big) \cdot \tilde{\mathbf{R}}_{\text{len}}(\mathcal{S}), \tag{40}$$

where $\mathbf{R}_{\text{base}} = \mathbf{R}_{\text{format}} + \mathbf{R}_{\text{non-repeat}} + \mathbf{R}_{\text{acc}}$, and $\tilde{\mathbf{R}}_{\text{len}}$ is the conditional length-based self-reward.

Accordingly, the optimization objective becomes

$$\mathcal{L}_{\text{len}}(\theta) = \mathbb{E}_{\mathcal{S} \sim \pi_\theta}\big[\mathbf{R}_{\text{total}}(\mathcal{S})\big]. \tag{41}$$

**Policy-gradient form.** Using the score-function identity, the gradient is

$$\nabla_\theta \mathcal{L}_{\text{len}}(\theta) = \mathbb{E}_{\mathcal{S} \sim \pi_\theta}\big[\mathbf{R}_{\text{total}}(\mathcal{S}) \nabla_\theta \log \pi_\theta(\mathcal{S})\big]. \tag{42}$$

Compared with the additive reward decomposition, the proposed formulation introduces a multiplicative gating factor $\tilde{\mathbf{R}}_{\text{len}}$, which rescales the trajectory return according to reasoning efficiency.

**Effect of multiplicative length gating.** For notation simplicity, define

$$X(\mathcal{S}) = \mathbf{R}_{\text{base}}(\mathcal{S}) + \mathbf{R}_{\text{desc}}(\mathcal{S}), \qquad Y(\mathcal{S}) = \tilde{\mathbf{R}}_{\text{len}}(\mathcal{S}), \tag{43}$$

so that $\mathbf{R}_{\text{total}} = XY$. Then the policy gradient estimator can be written as

$$g(\mathcal{S}) = X(\mathcal{S})Y(\mathcal{S})\nabla_\theta \log \pi_\theta(\mathcal{S}). \tag{44}$$

Following the common approximation that the gradient variance is governed by the scalar return variance,

$$\text{Var}\big[g(\mathcal{S})\big] \approx \mathbb{E}\big[\|\nabla_\theta \log \pi_\theta(\mathcal{S})\|^2\big] \cdot \text{Var}[X(\mathcal{S})Y(\mathcal{S})]. \tag{45}$$

Therefore, the key question is how the multiplicative length reward affects $\text{Var}[XY]$.

Using the identity

$$\text{Var}[XY] = \mathbb{E}[X^2 Y^2] - \mathbb{E}[XY]^2, \tag{46}$$

and the fact that the conditional length reward is clipped into $[0, 1]$, i.e.,

$$0 \leq Y(\mathcal{S}) \leq 1, \tag{47}$$

we obtain

$$\mathbb{E}[X^2 Y^2] \leq \mathbb{E}[X^2], \tag{48}$$

which implies that the second moment of the gated return is upper bounded by that of the ungated return. Hence, the multiplicative design does not amplify the reward magnitude and prevents unbounded variance growth. In contrast, an additive length penalty may change both the scale and sign of the return more abruptly.

**Variance decomposition of the gated return.** Let $\mu_Y = \mathbb{E}[Y]$. A first-order decomposition around $\mu_Y$ gives

$$XY = X\mu_Y + X(Y - \mu_Y). \tag{49}$$

Thus,

$$\text{Var}[XY] = \mu_Y^2 \text{Var}[X] + \text{Var}\big[X(Y - \mu_Y)\big] + 2 \text{Cov}\big(X\mu_Y, \ X(Y - \mu_Y)\big). \tag{50}$$

This expression shows that the variance of the gated return consists of a rescaled base term $\mu_Y^2 \text{Var}[X]$ and an additional fluctuation term caused by the randomness of $Y$. Since $Y \in [0, 1]$, the fluctuation introduced by the length reward is inherently bounded. Therefore, the variance increase due to length-based self-reward is controlled, rather than arbitrary.

Moreover, when $Y$ is positively aligned with trajectory quality—that is, concise reasoning tends to co-occur with correct and well-structured outputs—the gating term suppresses inefficient trajectories while preserving high-quality ones. In this case, the effective variance over useful trajectories can even decrease, because the reward mass becomes more concentrated on trajectories with both correctness and concise reasoning.

**Role of the conditional activation.** The conditional definition

$$\tilde{\mathbf{R}}_{\text{len}} = \begin{cases} \mathbf{R}_{\text{len}}, & \text{if } \exists i, \ \mathbf{R}_{\text{acc}}^{(i)} > 0, \\ 1, & \text{otherwise}, \end{cases} \tag{51}$$

is important for stability. In the early training stage, when all rollouts fail to locate the correct target, directly penalizing reasoning length may suppress exploration and bias the policy toward trivially short but uninformative responses. By setting $\tilde{\mathbf{R}}_{\text{len}} = 1$ in this case, the optimization reduces to the ungated reward $X$, avoiding premature compression before the model has acquired basic task competence. Hence, the length reward is only activated on informative samples, which reduces harmful variance caused by noisy failure trajectories.

**Why the anchor $\mathcal{N}_0$ is necessary.** The comparative term $\mathbb{I}[\mathcal{N}^2 < \mathcal{N}^1]$ alone only enforces relative shortening between the two passes. Without an absolute anchor, the policy may exploit this reward by increasing both $\mathcal{N}^1$ and $\mathcal{N}^2$ simultaneously while still maintaining $\mathcal{N}^2 < \mathcal{N}^1$. This leads to reward hacking and unstable optimization. The anchor-based penalty

$$\gamma \max(0, \mathcal{N}^1 - \mathcal{N}_0) \tag{52}$$

introduces an absolute reference for the first-pass reasoning length, discouraging pathological length inflation. Therefore, $\mathcal{N}_0$ regularizes the reward landscape and makes the optimization target better behaved.

**Optimization interpretation.** Overall, the proposed reward can be interpreted as a bounded, conditional reweighting of the original return:

$$\mathbf{R}_{\text{total}} = X \cdot Y, \qquad 0 \leq Y \leq 1. \tag{53}$$

This design preserves the correctness-oriented objective while encouraging concise reasoning only when the policy already exhibits task-relevant signal. As a result, the optimization balances two goals: (1) maintaining answer accuracy and structured outputs through $\mathbf{R}_{\text{base}} + \mathbf{R}_{\text{desc}}$, and (2) improving reasoning efficiency through a bounded multiplicative gate. This explains why the proposed method can shorten reasoning chains without causing severe instability in policy-gradient training.

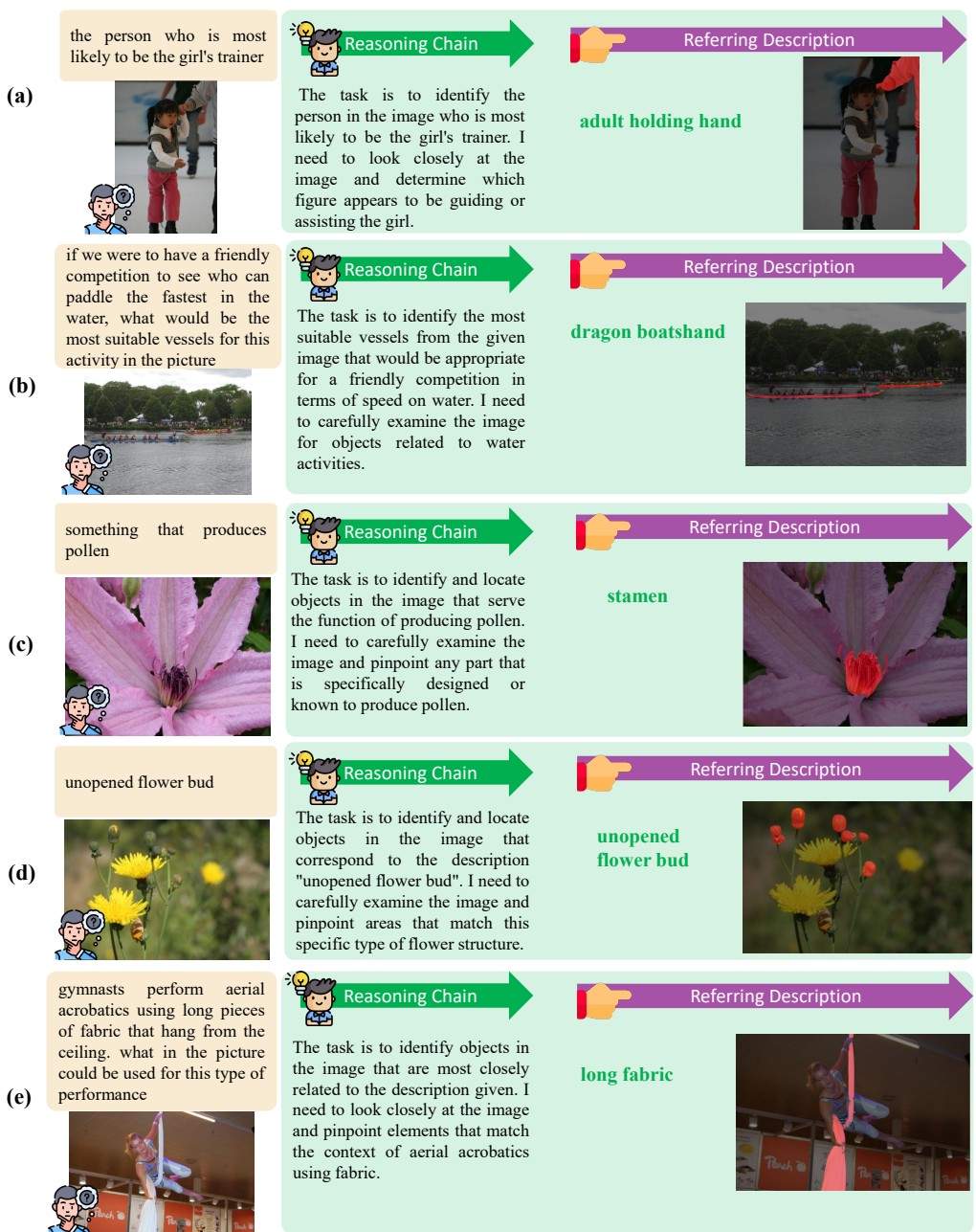

*Figure 5.* **Quantitative analysis of thinking and descriptive content generation in DR$^2$Seg.**

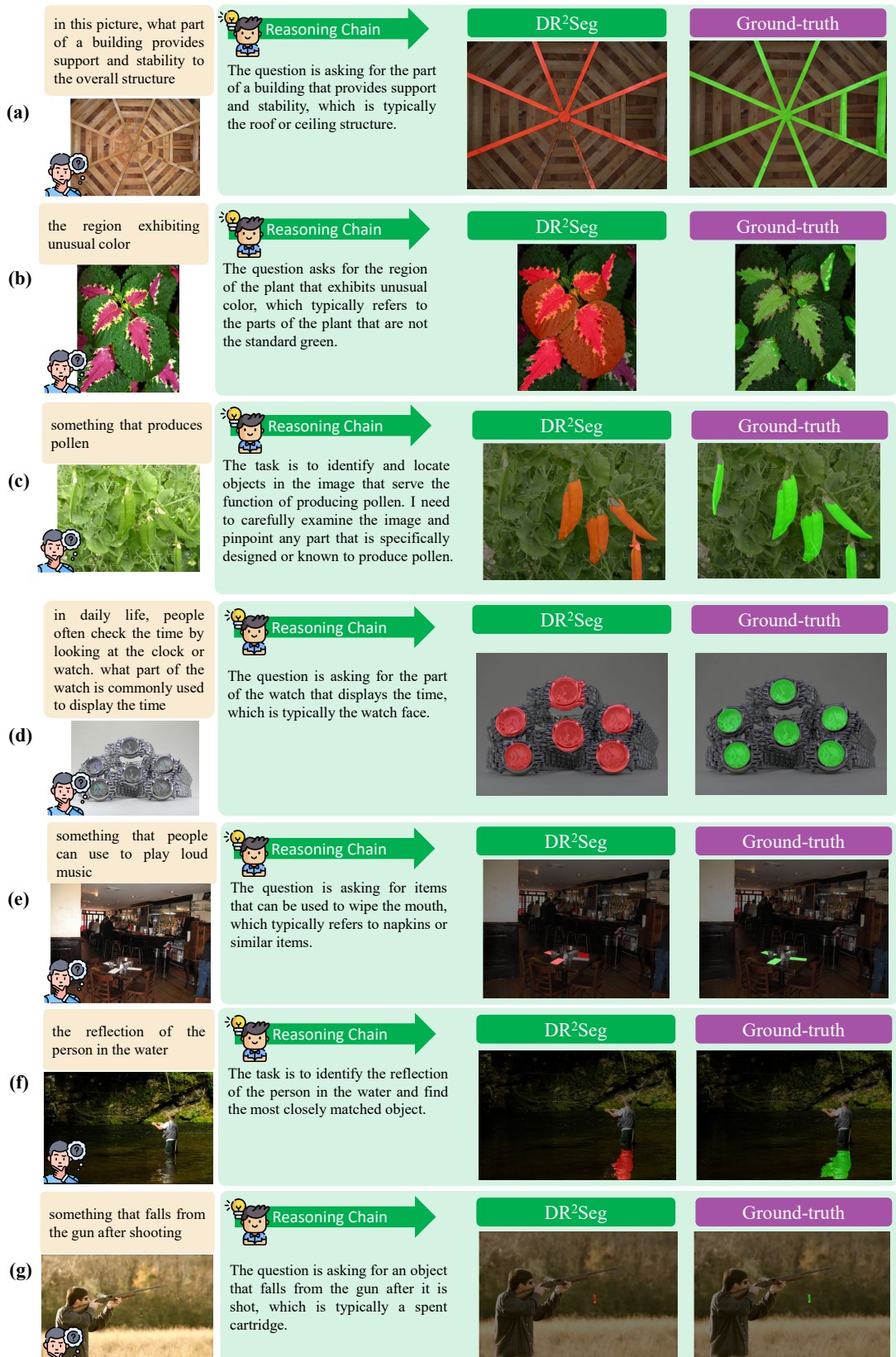

*Figure 6.* **More quantitative resuls of DR$^2$Seg in varying scenarios.** (a) Irregular objects; (b) Multi-target local regions; (c) Subtle camouflage; (d, e) Multi-object scenes; (f) Reflection/mirror scenarios; (g) Small-object cases.

