# OpenReview forum: "DR$^2$Seg: Decomposed Two-Stage Rollouts for Efficient Reasoning Segmentation in Multimodal Large Language Models"
_ICML.cc/2026/Conference — ICML 2026 regular_

### Official Review · Reviewer_yxDy · 2026-02-28

**Soundness:** 3
**Presentation:** 3
**Significance:** 3
**Originality:** 3
**Overall Recommendation:** 5
**Confidence:** 3

**Summary:**

------

To address the problem of overthinking in multimodal large models for reasoning segmentation, this paper propose a two-stage self-reward reinforcement learning framework. The core idea is to explicitly decompose reasoning segmentation into two sub-tasks: (1) multimodal reasoning, which generates an explicit target description, and (2) referring segmentation based on the generated description.

------

**Compliance With Llm Reviewing Policy:**

Affirmed.

**Final Justification:**

After the rebuttal, all of my original concerns were successfully addressed, so I decided to increase my rating.

**Key Questions For Authors:**

------

1. Please refer to the concerns raised in the Weaknesses section.

2. The overall presentation still requires further polishing. For example, in Figure 2, the arrow from “description” to “answer” is misaligned, and the yellow box exceeds the gray background boundary. In Figure 3, the purple and green background boxes are not properly aligned. There are several minor issues in the figures that should be carefully revised to improve visual quality and professionalism.

3. The formulas in the appendix require more detailed explanations, particularly in the section on Group-Relative Policy Optimization.

------

**Limitations:**

yes

**Strengths And Weaknesses:**

------

**Strengths**

1. The paper identifies a critical issue in RL-based reasoning segmentation, namely that overthinking leads to attention diffusion and degraded performance.

2. The proposed two-stage rollout design is intuitive, and the reward construction is reasonable. The method is supported by a certain level of theoretical analysis.

3. The experimental evaluation is relatively comprehensive. The approach achieves state-of-the-art performance on the ReasonSeg and RefCOCO series datasets, while significantly reducing the number of inference tokens.

---


**Weaknesses**

1. Compared with more theoretically grounded RL frameworks, the proposed method still appears largely heuristic, and the theoretical justification remains somewhat conceptual.

2. It is unclear whether the proposed reasoning–segmentation “decoupling” truly holds in practice. More direct empirical evidence would strengthen this claim.

3. It remains questionable whether shorter reasoning is always better. How does the method distinguish between overthinking and necessary reasoning? Could it negatively affect essential chain-of-thought processes? Are there potential corner cases where performance degrades?

4. Can the change in mutual information be quantitatively measured?

------

---

> ### Author Rebuttal · Authors · 2026-03-30
>
> **Thanks for your careful review.** We appreciate your recognition that our work identifies a critical issue, presents a sound design with theoretical support, and achieves SOTA performance with fewer inference tokens.
>
> ### **Q1: Methodology and theory:**
>
> To clarify, the methodology is **intentionally presented in a simple and intuitive way for readability**, but the design principles are not merely heuristics. Our method is supported from both theoretical and empirical perspectives. In Sec. 3.4, we provide both **Optimization Analysis** and **Information-Theoretic Analysis**, with detailed derivations in the Supplementary Material Sec. C. More broadly, self-rewarding RL designs in related work (e.g., Calibrated Self-Rewarding[NeruIPS'24] and Vision-SR1[ICLR'26]) are also often introduced in an intuitive form for clarity, while still being grounded in theoretical analysis.
>
> Specifically, from an optimization perspective, the analysis explains how the self-rewarding objective steers training toward more target-relevant intermediate descriptions, thereby improving credit assignment. From an information-theoretic perspective, the explicit description bottleneck helps filter irrelevant reasoning while retaining the information for downstream grounding.
>
> Empirically, our design is also supported by both quantitative and qualitative evidence. The quantitative results in Tab. 1 show consistent improvements in both accuracy and efficiency over the baseline, while the qualitative examples in Fig. 3 illustrate examples. Moreover, Fig. 4 offers empirical evidence from an information-theoretic perspective through answer entropy.
>
> We will further polish the theoretical analysis.
>
> ### **Q2: More empirical evidence:**
>
> To provide more direct empirical evidence, we further evaluate the second pass using only the generated description $\mathcal{D}$ as input, without the original question query $\mathcal{Q}$. The results are as follows:
>
> | Method                              |     gIoU |     cIoU |
> | ----------------------------------- | -------: | -------: |
> | VisionReasoner |     62.3 |     54.3 |
> | $\mathcal{D}$-only (Second-Pass)    |     65.9 |     62.6 |
> | DR$^2$Seg                           | **66.1** | **63.6** |
>
> First, introducing the description together with the two-stage rollout yields clear improvements over one-stage rollout (VisionReasoner). Second, even when the second-pass takes only the description $\mathcal{D}$ as input, it still achieves competitive performance. This suggests that $\mathcal{D}$ preserves most of the target-relevant information for downstream grounding.
>
> Fig. 5 in the supplementary material shows that even with very few description tokens, the model can still generate accurate and unambiguous descriptions, such as *adult holding hand*.
>
> Furthermore, we conducted validation on multi-target reasoning segmentation, demonstrating that our method can generalize naturally to more complex scenarios. Please refer to Reviewer nB7J Q3 for further details.
>
> We will include the above analysis in the revised version.
>
> ### **Q3: Overthinking and necessary thinking:**
>
> We’d like to clarify that our method **does not assume that shorter reasoning is always better**. Instead, it encourages shorter reasoning only when the model can generate correct answers. In other words, our goal is to suppress **redundant** reasoning rather than necessary reasoning.
>
> Specifically, in Eq. 8, the length-based reward is activated only when at least one rollout receives a positive accuracy reward; otherwise, it is disabled. This avoids imposing overly pressure toward shorter reasoning before successful target localization, while preserving exploration on difficult examples. Therefore, model is encouraged to produce reasoning that is **concise yet sufficient**, rather than merely short.
>
> Regarding the systematic analysis of failure cases, please refer to Reviewer 79KA Q2.
>
> ### **Q4: Quantify mutual information:**
>
> We initially attempted to quantify mutual information, but this is nontrivial in practice due to the varying latent reasoning state. In the current paper, instead of directly estimating the full mutual information, we use the entropy of the segmentation answer tokens in Fig. 4 as a practical quantitative proxy for localization uncertainty. Importantly, this entropy is computed only over the final answer tokens, excluding reasoning tokens, so it specifically reflects the model’s confidence in localization rather than variability in the reasoning trace itself. As shown in Fig. 4, the answer entropy consistently decreases while accuracy improves during training, suggesting that the two-stage rollout leads to less ambiguous and more confident grounding decisions.
>
> ### **Q5: Some typos:**
>
> We will carefully correct these typos in the revised manuscript.
>
> ### **Q6: More explanation of derivations:**
>
> We will expand the derivations and provide additional step-by-step clarification to improve readability.

---

> > ### Author Rebuttal · Reviewer_yxDy · 2026-04-01
> >
> > - **About Weakness 3 (It remains questionable whether shorter reasoning is always better. How does the method distinguish between overthinking and necessary reasoning? Could it negatively affect essential chain-of-thought processes? Are there potential corner cases where performance degrades?).**
> >
> >    - Could you provide more quantitative analysis?

---

> > > ### Author Response · Authors · 2026-04-01
> > >
> > > Thank you very much for your timely response and for your interest in our work. We are pleased that our previous response has addressed most of your concerns. We further refine our analysis of failure cases from **overthinking** and **insufficient thinking**.
> > >
> > > In fact, in the bottom-left of Fig. 1 in the main paper, we have already quantified failure cases from the decoupled perspectives of **reasoning errors** and **segmentation errors**. Nevertheless, we also **agree that categorizing failures into overthinking and insufficient thinking allows for a more fine-grained analysis of the model’s capability in reasoning**.
> > >
> > > We first define the criteria for overthinking and insufficient thinking as follows:
> > >
> > > * **Overthinking**: the reasoning correctly identifies the target, but the segmentation object is clearly incorrect. We exclude cases where the MLLM localizes the target correctly but the failure is caused solely by SAM’s segmentation.
> > > * **Insufficient thinking**: the reasoning fails to derive the correct target. Since some failures may stem from the model’s intrinsic knowledge limitations (see Reviewer 79KA, Q2 on *Knowledge Deficiency*), we only count cases where VisionReasoner succeeds, but DR$^2$Seg fails because it does not generate the correct description.
> > >
> > > On the **ReasonSeg-Val** set, VisionReasoner has 32 failure cases in total, among which 14 are overthinking cases, accounting for **43.7%** (14/32). For DR$^2$Seg, 12 of these 14 overthinking cases are corrected (**85.7%**), while 3 insufficient-thinking cases are introduced, accounting for **12.5%** of its 24 total failures (3/24).
> > >
> > > On the **ReasonSeg-Test** set, VisionReasoner has 112 failure cases in total, among which 35 are overthinking cases, accounting for **31.2%** (35/112). For DR$^2$Seg, 27 of these 35 overthinking cases are corrected (**77.1%**), while 10 insufficient-thinking cases are introduced, accounting for **10.6%** of its 94 total failures (10/94).
> > >
> > > These results show that overthinking accounts for a substantial portion of the failures in VisionReasoner, and DR$^2$Seg can **correct most of these overthinking cases**. However, although DR$^2$Seg adopts the gating design in Eq. 8 of the main paper to determine whether reasoning should be compressed based on the accuracy reward during training, it still introduces a small number of insufficient-thinking failures on the test set.
> > >
> > > To further validate the effectiveness of our method in more complex scenarios, we additionally evaluate it on **the MUSE benchmark**, where a single query may involve multiple semantically distinct targets within the reasoning process. Please see Reviewer nB7J, Q3. The results show that our method **can naturally adjust both the reasoning and the description according to the task difficulty**.

---

### Official Review · Reviewer_79KA · 2026-03-07

**Soundness:** 3
**Presentation:** 3
**Significance:** 3
**Originality:** 3
**Overall Recommendation:** 4
**Confidence:** 4

**Summary:**

This paper introduces a novel framework designed to improve the efficiency and accuracy of reasoning segmentation in Multimodal Large Language Models (MLLMs). The core problem addressed is "overthinking"—where MLLMs generate excessively long and redundant reasoning chains that not only waste computation but also confuse the subsequent object localization step, degrading performance.

Experiments on the ReasonSeg benchmark and referring expression datasets show that DR2Se significantly outperforms previous state-of-the-art methods like VisionReasoner, achieving higher segmentation accuracy while using substantially fewer reasoning tokens. The paper includes thorough ablations, theoretical analysis , and demonstrations of generalizability across different MLLM scales and segmentation models.

**Compliance With Llm Reviewing Policy:**

Affirmed.

**Key Questions For Authors:**

See the weaknesses.

**Limitations:**

See the weaknesses.

**Strengths And Weaknesses:**

Strengths:

S1: The paper clearly identifies the "overthinking" problem in reasoning segmentation, where verbose reasoning degrades performance. The proposed two-stage rollout is a simple yet powerful idea that directly addresses this by decoupling complex reasoning from perception. The self-reward mechanism derived from this structure is elegant, as it leverages the model's own consistency to provide supervision, avoiding the need for external, larger models  and making the approach more self-contained and fair.

S2: The paper is well-structured and clearly written. The introduction effectively motivates the problem by contrasting the proposed approach with existing methods . Figure 1 provides a clear, high-level overview of the core idea. The methodology section is detailed, and the self-reward design principles are explicitly stated and linked to the two-stage rollout, making the logic easy to follow.

Weaknesses:

W1: While the self-reward mechanism is novel in this context, the core idea of using a generated description as an intermediate bottleneck has parallels in some prior vision-language work. The paper could strengthen its novelty argument by more clearly distinguishing its contribution from these related, but not identical, concepts.

W2:  The paper presents many successful qualitative examples (Figures 3 and 6) but does not include a systematic analysis of failure cases. For a method that relies on the quality of the generated description D, it would be insightful to understand when and why the second pass fails.

---

> ### Author Rebuttal · Authors · 2026-03-30
>
> **Thank you very much for your positive evaluation of our paper.** We especially appreciate your recognition that our method is built on a simple yet powerful idea, features an elegant derived reward design, and is presented in a clear and well-structured manner. We will now address the following concerns in detail.
>
> ### **Q1: Distinguishing our contribution from related self-rewarding vision-language work**
>
> We appreciate this constructive suggestion. As noted in Lines 139-143 of the main paper, *“This work further explores self-rewarding for reasoning segmentation, focusing on pixel-level object understanding through decomposition into reasoning and referring segmentation via an explicit description bottleneck.”* We will make this distinction clearer in the revised manuscript.
>
> Specifically, unlike most prior vision-language methods that rely on textual descriptions for Visual Question Answering (VQA), **reasoning segmentation relies more heavily on fine-grained visual details, making models more vulnerable to attention distraction caused by redundant or irrelevant preceding reasoning steps**. Our self-rewarding design explicitly decouples reasoning from perception, which is particularly effective in alleviating this issue.
>
> Moreover, the role of the description in our method differs substantially from that in prior VQA-style work. In VQA, descriptions are mainly introduced to supplement visual information and prevent the model from bypassing the image and relying solely on textual reasoning, which may lead to hallucination. In contrast, in our method, **the description serves as an intermediate signal that transforms a complex reasoning query into a referring-style instruction that is easier to localize visually**. The self-reward mechanism further provides effective supervision, encouraging the generated description to stay closer to the target object and thereby better support downstream perception.
>
> ### **Q2: Systematic analysis of failure cases**
>
> We thank the reviewer for this valuable suggestion. We further visualize representative failure cases during training.
>
> The visualization results are available at [[Anno. Link]](https://anonymous.4open.science/api/repo/failure_visual-3D93/file/failure_visual.pdf?v=9d53d1f4).
>
> Based on our qualitative analysis, we observe three typical failure modes:
>
> 1. **Knowledge deficiency**, where the model lacks the necessary world knowledge or commonsense and thus generates an incorrect description (e.g., not knowing that the black 8 ball is the final target in pool).
>
> 2. **Coarse description**, where key discriminative parts or attributes are omitted.
>
> 3. **Ambiguous description**, where multiple similar instances satisfy the generated description.
>
> The first type mainly reflects a limitation of the base model and may require external knowledge augmentation (e.g., RAG), while the latter two are more likely to be alleviated through further training.
>
> To further assess the quality of the generated description $\mathcal{D}$, we evaluate a second pass using only $\mathcal{D}$ as input on ReasonSeg-test. The results are as follows:
>
> | Method                           |     gIoU |     cIoU |
> | -------------------------------- | -------: | -------: |
> | $\mathcal{D}$-only (Second-Pass) |     65.9 |     62.6 |
> | DR$^2$Seg                        | **66.1** | **63.6** |
>
> Notably, when running the second pass using only the generated description $\mathcal{D}$, without the original query, the accuracy remains competitive, suggesting that $\mathcal{D}$ already captures most of the target-relevant information.
>
> Consistently, Figure 5 in the supplementary material shows that even with few description tokens, the model can still produce accurate and unambiguous referring descriptions, such as *adult holding hand* and *unopened flower bud*.
>
> We will include representative failure cases and a more systematic discussion in the revised manuscript.

---

> > ### Author Rebuttal · Reviewer_79KA · 2026-04-05
> >
> > Thanks for your responses. No more questions.

---

> > > ### Author Response · Authors · 2026-04-06
> > >
> > > Dear Reviewer 79KA:
> > >
> > > Thank you very much for your time and for acknowledging our efforts during the rebuttal. We are very glad to hear that our responses have well addressed your concerns.
> > >
> > > Sincerely,
> > >
> > > Authors of Submission 3042

---

### Official Review · Reviewer_nB7J · 2026-03-09

**Soundness:** 2
**Presentation:** 3
**Significance:** 2
**Originality:** 3
**Overall Recommendation:** 4
**Confidence:** 3

**Summary:**

The paper addresses the issue of overthinking in Multimodal Large Language Models (MLLMs) applied to reasoning segmentation tasks. The authors propose DR2Seg, a self-rewarding reinforcement learning framework. The method employs a two-stage rollout strategy during training: in the first stage, the model generates an explicit target description from a complex query ; in the second stage, this description replaces the original query to verify if it is self-contained enough to produce the correct segmentation. Experiments are on the ReasonSeg and RefCOCO datasets using Qwen2.5-VL and SAM variants.

**Compliance With Llm Reviewing Policy:**

Affirmed.

**Final Justification:**

In the final rebuttal respone, the authors have addressed most of my concerns.

**Key Questions For Authors:**

1. How do you select the length anchor $\mathcal{N}_{0}$ for new datasets or different base MLLMs? Is there a principled approach to setting this parameter without exhaustive hyperparameter tuning?
2. Can the two-stage rollout method easily extend to complex, multi-target reasoning segmentation where a single query might require segmenting multiple distinct objects that cannot be summarized in a simple description?
3. Could you provide a plot showing the reward changes according to different values of $\mathcal{N}_{0}$ and $\gamma$? The current reward curve dynamics are not easy to grasp, and visualizing how these parameters affect the reward over time would be helpful.

**Limitations:**

yes

**Strengths And Weaknesses:**

**Strength**:

1. The paper is clearly written, well-organized, and easy to follow.
2. The authors provide a robust and comprehensive set of experiments and ablation studies, effectively validating the various components of their framework.
3. The proposed method achieves strong empirical results on both the ReasonSeg benchmark and RefCOCO datasets.
4. The algorithm design is highly effective: it achieves performance gains purely through self-rewarding mechanisms without relying on external supervision.

**Weakness**:
1. The framework exhibits sensitivity to the predefined length anchor hyperparameter ($\mathcal{N}_{0}$). As shown in the ablations table 4, poorly chosen values can drastically affect the model's ability to infer targets and control reasoning length.
2. I would also concern regarding the constraints on generalization.
    - The authors mentioned in both introduction and related work that the sft-based methods lack generalization ability, but did not compare with them in the experiment.
    - The authors only evaluate the models on the data that is aligned with the training set: RefCOCO has the similar distribution with part of the training set of VisionReasoner, and ReasonSeg val/test has similar distribution of ReasonSeg train.

   This leads to the unforeseen generalization ability, which contradicts to the motivation.
3. The guiding principles proposed in the methodology feel somewhat heuristic and lack solid empirical grounding to fully justify their correctness across diverse scenarios. Additional qualitative results demonstrating these principles in action would be beneficial.
4. The theoretical analysis section  lacks rigor. Specifically, in Section 3.4, the authors state that the length-based reward is excluded to isolate structural contributions. However, this exclusion hurt the analysis because the length reward is applied multiplicatively , whereas the other rewards are affine combinations. This multiplicative nature fundamentally influences convergence, bias, and variance, and thus must be the main point of the analysis. Furthermore, the claim that $R_{desc}$ improves credit assignment is presented as an intuitive statement without rigorous mathematical proof. The optimization analysis relies heavily on loose intuition rather than formal proofs, and the main text's theoretical claims seem disconnected from the detailed derivations provided in Appendix B.1.

---

> ### Author Rebuttal · Authors · 2026-03-30
>
> **Thanks for your careful review.** We appreciate your recognition of our paper’s clarity, robust experimental support, strong empirical results, and effective algorithmic design.
>
> ### **Q1: Choose suitable $N_0$:**
>
> We’d like to claim that the results under different $N_0$ are **reasonable**. As shown in Tab. 4, when $N_0$ decreases from 55 to 45, reasoning becomes shorter while accuracy improves, suggesting that the original reasoning contains redundancy. Reducing $N_0$ from 45 to 35 further compresses reasoning, but slightly hurts accuracy due to over-compression. When $N_0$ is too small (e.g., 25), the model cannot reliably infer the target, leading to a conflict between the accuracy and length rewards. Overall, this trend is clear and interpretable.
>
> For new datasets or MLLMs, a practical strategy is to first measure the average reasoning token without the length-based self-reward (e.g., 46 tokens in DR²Seg), and then choose candidate $N_0$ values around that average. In practice, a suitable $N_0$ can be identified within only 3-4 trials.
>
> ### **Q2: Regarding generalization:**
>
> We respectfully clarify that **generalization is also a key consideration in our work**.
>
> First, Tab. 1 already includes SFT baselines (e.g., LISA and CORES). To further address your concern, we compare **SFT vs. RL under the same MLLM** (Qwen2.5-VL-3B) on both in-distribution and out-of-distribution benchmarks:
>
> | Method | RefCOCOg (ID) | ReasonSeg (OOD) |
> | ------ | ------------: | --------------: |
> | SFT    |          70.8 |            44.9 |
> | RL     |          **73.6** |            **53.8**|
>
> This shows that RL largely improves OOD generalization.
>
> Second, our experiments also validate the generalizability. In the **zero-shot** setting, DR²Seg is trained on the same distribution as VisionReasoner (aligned with RefCOCO) and directly evaluated on ReasonSeg. DR²Seg improves gIoU by +1.2% (Tab. 1). In the **few-shot** setting, after incorporating ReasonSeg-train, DR$^2$Seg improves further on ReasonSeg while retaining gains on RefCOCO (Tab. 2), e.g., +1.9% on RefCOCOg. These **cross-dataset** results demonstrate its generalizable reasoning and grounding ability.
>
> ### **Q3: Diverse scenarios:**
>
> Regarding the theoretical analysis, please see Reviewer yxDy Q1.
>
> To validate our method in diverse scenarios, we further evaluate it on **MUSE** [1] benchmark, where a single query may involve multiple semantically distinct targets within the reasoning process. The results show that DR²Seg generalizes well to multi-target scenarios.
>
> | Method         | MUSE-val |      | MUSE-test |      |
> | -------------- | -------: | ---: | --------: | ---: |
> |                |     gIoU | cIoU |      gIoU | cIoU |
> | VisionReasoner |     51.7 | 48.6 |      51.1 | 46.4 |
> | DR²Seg (Ours)      |     **55.7** | **54.5** |      **54.7** | **50.8** |
>
> We also provide qualitative samples in [[Anon. Link]](https://anonymous.4open.science/api/repo/MUSE_Visualization-3653/file/MUSE_Visualization.pdf?v=dee98801), showing that DR²Seg can generate descriptions that include multiple relevant objects. We will include these experiments.
>
> [1] Pixellm: Pixel reasoning with large multimodal model. CVPR 2024.
>
> ### **Q4: Length-based reward analysis:**
>
> We exclude the length-based reward to isolate the **structural contribution** of the two-stage rollout, which is the foundation of this work. To further clarify the role of the length-based reward, let
>
> $
> R_b = R_{\mathrm{base}} + R_{\mathrm{desc}}, \quad
> R_{\mathrm{total}} = R_b \cdot \widetilde{R}_{\mathrm{len}},
> $
>
> where $\widetilde{R}_{\mathrm{len}} \in [0,1]$ and is disabled when accuracy is zero. Then
>
> $
> \nabla \mathcal{L} \approx \mathbb{E}\left[R_{\mathrm{total}}\nabla \log \pi\right]
> = \mathbb{E}\left[R_b \cdot \widetilde{R}_{\mathrm{len}} \nabla \log \pi\right].
> $
>
> This multiplicative form has two effects:
>
> (1) **Reward shaping toward concise reasoning**, penalizing unnecessarily long chains once sufficient information is captured;
>
> (2) **Preserving exploration** early on, since $\widetilde{R}_{\mathrm{len}}=1$ when no correct answer is found.
>
> Moreover, Fig. 4 provides empirical support for the improved credit assignment effect through **answer entropy**, showing reduced uncertainty in answer generation. The main text provides the core intuition behind the theory, while Appendix B.1 contains the formal derivations.
>
> We will add a detailed analysis of $ {R}_{\mathrm{len}}$ and explanation of derivations.
>
> ### **Q5: Reward changes with $N_0$ and $\lambda$:**
>
> The reward-change curves are provided in [[Anon. Link]](https://anonymous.4open.science/api/repo/Reward-Change_Curves-4BAD/file/Reward-Change_Curves.pdf?v=b9a8fbf3).
>
> As $N_0$ decreases, it becomes more challenging for the model to learn to follow Principle 2 in the paper. However, once the model learns to adhere to this principle, the reward becomes stable. For $\lambda$, the reward varies only slightly across settings, consistent with the results in Tab. 5.

---

> > ### Author Rebuttal · Reviewer_nB7J · 2026-04-03
> >
> > 1. Q1: while I understand the reason length and the accuracy change is reasonable, my concern is more about the standalone performance when it is set to 45. This requires an accurate estimation of $\mathcal{N}_0$  at every experimental setting. If you set $\mathcal{N}_0$ to be the average length without length reward, would it have high randomness? Do you need to perform iterative experiments to find better $\mathcal{N}_0$? How do your current experiments define this hyper-parameter? I believe the discussion and analyses of such setting should be more comprehensive.
> > 2.  Q4: Could you provide variance analysis (such as equation 28) for your defined policy gradient?

---

> > > ### Author Response · Authors · 2026-04-04
> > >
> > > Thank you very much for your timely response and your interest in our work. We are pleased that our previous response has addressed most of your concerns.
> > >
> > > ### **Q1: More in-depth analysis about $N_0$**:
> > >
> > > In our experiments, $N_0$ is defined in a simple and fixed manner rather than being obtained through iterative tuning. Specifically, we first measure the **average reasoning length of  DR²Seg without the length reward under the zero-shot setting**, which is approximately **46 tokens** (Tab. 1). Based on this anchor, we sample **55, 45, 35, and 25** as nearby values for the ablation study in Tab. 4. Once selected before training, $N_0$ remains **fixed throughout the training process**.
> > >
> > > Regarding randomness, the average reasoning length is a **dataset-level statistic** rather than a single-sample measurement. Its variation across dataset splits is overall controllable. Under the setting without the length reward, the token-length differences between val and test sets are 0.1 for Seg-Zero, 0.7 for PixelThink, 4.0 for VisionReasoner, and 9.2 for  DR²Seg (Tab. 1). Although some variation exists, it is still within a controllable range and supports local sampling without requiring large-scale retuning.
> > >
> > > Inspired by the reviewer’s **valuable suggestion**, we further evaluate an **adaptive variant** that iteratively re-estimates the average reasoning length and updates $N_0$ during training. However, this strategy requires more training time, as the model need to continuously adapt to the dynamically updated $N_0$. Specifically, we first train the model for 50 steps of pre-training to allow it to follow Principle 2 in the paper. Then, every epoch (13 steps), the model re-estimates $N_0$ for the next stage. The results of this iterative training process on ReasonSeg-val are shown below.
> > >
> > > | Step   |    0 |   50 |   63 |   76 |   89 |  102 |      115 |  128 |  141 |
> > > | ------ | ---: | ---: | ---: | ---: | ---: | ---: | -------: | ---: | ---: |
> > > | $N_0$  |    - |   75 |   61 |   46 |   37 |   33 |       30 |   27 |   27 |
> > > | gIoU   | 66.0 | 67.2 | 68.1 | 68.8 | 67.8 | 67.7 | **68.5** | 68.1 | 67.7 |
> > > | Tokens | 75.2 | 61.0 | 46.2 | 37.4 | 33.7 | 30.8 | **27.9** | 25.9 | 25.2 |
> > >
> > > We also report the reward-change curves of this adaptive variant in [[Anno. Link]](https://anonymous.4open.science/api/repo/N_0_Adaptive_Reward-DD78/file/N_0_Adaptive_Reward.pdf?v=61b74a94).
> > >
> > > These results show that once $N_0$ is reduced beyond a certain point, the reasoning tokens become hard to further decrease, indicating that the compression has reached saturation. Moreover, the overall best trade-off between accuracy (68.5 gIoU) and efficiency (27.9 Tokens) corresponds to the inflection point. This best result is similar to the result obtained by directly setting $N_0$ to 45 (Tab. 4). This adaptive manner eliminates the need for the sampling of $N_0$, but it requires more training time (~3 $\times$). The reward-change curves also show that every adjustment to $N_0$ introduces reward fluctuations, which ultimately stabilize in the later stages.
> > >
> > > We will include the results of this adaptive variant in Tab. 4 of the main paper, along with additional discussion to further clarify the selection of $N_0$.
> > >
> > >
> > >
> > > ### **Q2: Variance analysis of length-based reward**:
> > >
> > > Here, let
> > > $
> > > X = \mathbf R_{\text{base}} + \mathbf R_{\text{desc}},
> > > \qquad
> > > Y = \tilde{\mathbf R}_{\text{len}} \in [0,1].
> > > $
> > >
> > > Then the total reward is
> > > $
> > > \mathbf R_{\text{total}} = XY,
> > > $
> > >
> > > and the corresponding policy-gradient estimator is
> > > $
> > > g(\mathcal S) = XY \nabla_\theta \log \pi_\theta(\mathcal S).
> > > $
> > >
> > > Under the standard approximation,
> > > $
> > > \operatorname{Var}[g(\mathcal S)]
> > > \approx
> > > \mathbb E\left[\left||\nabla_\theta \log \pi_\theta(\mathcal S)\right||^2\right]
> > > \operatorname{Var}[XY],
> > > $
> > > the central quantity is therefore ($\operatorname{Var}[XY]$).
> > >
> > > Because (Y $\in$ [0,1]), we have
> > > $
> > > |XY| \le |X|,
> > > \qquad
> > > \mathbb E[X^2Y^2] \le \mathbb E[X^2].
> > > $
> > >
> > > This implies that the multiplicative length-based reward does not amplify the return magnitude and does not cause unbounded growth of the second moment.
> > >
> > > Moreover, letting $\mu_Y = \mathbb E[Y]$, we can decompose
> > > $
> > > XY = \mu_Y X + X(Y-\mu_Y),
> > > $
> > > which gives
> > >
> > > $\operatorname{Var}[XY]=
> > > \mu_Y^2 \operatorname{Var}[X]
> > > +
> > > \operatorname{Var}\big(X(Y-\mu_Y)\big)
> > > +
> > > 2\operatorname{Cov}\big(\mu_Y X, X(Y-\mu_Y)\big).
> > > $
> > >
> > > Compared with the Eq. 28 in the main paper, the length-based reward affects optimization through a **bounded multiplicative gate**. This design suppresses excessively long reasoning trajectories without increasing reward scale. As a result, it provides a controlled way to encourage concise reasoning while keeping gradient variance well behaved.
> > >
> > > More importantly, the product form $XY$ makes accuracy reward and length reward **mutually constrained**: a trajectory can obtain a high total reward only when it performs well on both dimensions simultaneously. This leads to a joint optimization of correctness and conciseness.

---

### Official Review · Reviewer_qLaq · 2026-03-11

**Soundness:** 2
**Presentation:** 2
**Significance:** 2
**Originality:** 3
**Overall Recommendation:** 4
**Confidence:** 4

**Summary:**

This paper proposes DR2Seg, a self-rewarding RL framework for reasoning segmentation with multimodal LLMs. The key motivation is that existing RL-based reasoning segmentation methods tend to “overthink”: verbose chain-of-thought increases token cost and may diffuse attention, harming localization and downstream mask quality.  DR2Seg introduces a decomposed two-stage rollout: in the first stage, reasoning is used to make the implicit segmentation target more explicit, and in the second stage, the model performs referring segmentation. Experiments on Qwen2.5-VL (3B/7B) with SAM2/SAM3 across ReasonSeg and RefCOCO-style benchmarks show consistent reductions in reasoning tokens with improved or competitive segmentation accuracy.

**Compliance With Llm Reviewing Policy:**

Affirmed.

**Key Questions For Authors:**

None

**Strengths And Weaknesses:**

**Strengths**

1、The paper is clearly motivated: it leverages the reasoning capability of MLLMs to derive an explicit target for reasoning segmentation, and then performs standard referring segmentation.

2、Compared to approaches that inject difficulty estimates from very large auxiliary models, DR2Seg stays “self-contained” and avoids extra supervision/modules.

3、Extensive experiments validate the effectiveness and generalization of DR2Seg across MLLMs of varying scales (Qwen2.5-VL (3B/7B)) and segmentation models (SAM2/SAM3)

**Weaknesses**

1、 The proposed method is primarily designed to address reasoning segmentation. I acknowledge that the first-stage reasoning can make the segmentation target more explicit and reduce redundancy in natural language. However, the paper lacks both quantitative and qualitative evidence showing which specific samples benefit from this design—i.e., what types of failure cases in VisionReasoner are actually improved.

2、On line 184, the paper states, “Notably, only a single pass (i.e., the first pass) is required during inference.” Why is that? Isn’t the reasoning process supposed to be two-stage? does the performance gain induced by the second-pass self-verification actually transfer to the first-pass outputs?

3、While the paper emphasizes token reduction, the efficiency comparison may not be entirely apples-to-apples: different baselines can use different output formats, CoT constraints, and even decoding settings. Moreover, DR2Seg performs two rollouts during training but only reports inference-time tokens. It would be helpful to more clearly disentangle and report training compute, inference compute, and wall-clock time under matched decoding configurations to support the efficiency claims.

---

> ### Author Rebuttal · Authors · 2026-03-30
>
> **Thank you very much for your positive evaluation of our paper.** We appreciate your recognition of DR²Seg’s clear motivation, self-contained design, and its ability to avoid extra supervision. Additionally, we are glad that the effectiveness and generalization of DR²Seg across models has been validated. We will now address the following concerns in detail.
>
> ---
>
> ### **Q1: More quantitative and qualitative evidence for the design.**
>
> Failure cases in VisionReasoner, especially those caused by **overthinking-induced recognition confusion**, are alleviated in DR²Seg. For example, after correctly identifying the target, it may continue to overanalyze other irrelevant factors, which eventually interferes with the final prediction. In fact, we **have already** provided both quantitative and qualitative evidence in the manuscript.
>
> * **Quantitative Evidence**: As shown in the bottom-left corner of Figure 1, on the ReasonSeg Test benchmark, DR²Seg reduces reasoning errors by 11 cases (from 40 to 29) and segmentation errors by 7 cases (from 72 to 65) when compared to VisionReasoner. These results clearly demonstrate that our decoupled design leads to more target-oriented reasoning, which results in more accurate referring. Additionally, with more explicit referring descriptions, the LLM’s localization capability is also enhanced, contributing to fewer segmentation errors.
>
> * **Qualitative Evidence**: In Figure 3, we provide examples where VisionReasoner fails, but our method succeeds. These include both simple single-target scenarios (first and second rows) and more complex multi-target scenarios (third row). We present both the reasoning content and the corresponding segmentation results for each case.
>
> We will include more qualitative examples and a more detailed analysis in the revised version.
>
> ---
>
> ### **Q2: Why is only one pass needed?**
>
> To clarify, the DR²Seg method **requires only a single pass during inference**: (\$\mathcal{I}\$, \$\mathcal{Q}\$) \$\rightarrow\$ (\$\mathcal{R}\$, \$\mathcal{D}\$, \$\mathcal{A}\$). The two-stage rollout is only employed during training to learn the referring description \$\mathcal{D}\$.
>
> The second pass provides additional supervision for the description \$\mathcal{D}\$, acting as a training signal that further improves its quality. This improvement carries over to the first-pass output during inference, since the answer \$\mathcal{A}\$ is generated after the description \$\mathcal{D}\$. This explains why the performance gain from the second pass transfers to the single-pass inference. Therefore, our method **ensures a fair comparison** with other single-pass methods while highlighting its efficiency by alleviating overthinking.
>
> To address your question regarding whether the second-pass self-verification actually transfers to the first-pass outputs, we further evaluated a second-pass using only \$\mathcal{D}\$ as input on ReasonSeg-test. As shown below, the results were still promising even without the question query. This demonstrates that the learned description \$\mathcal{D}\$ is able to accurately refer to the target. We will include a more detailed analysis in the revised version.
>
> | Method                         | gIoU | cIoU |
> |--------------------------------|---------------------|---------------------|
> | $\mathcal{D}$-only (Second-Pass) | 65.9                | 62.6            |
> | DR²Seg | **66.1**            | **63.6**            |
> ---
>
> ### **Q3: How does DR²Seg compare in training vs. inference compute efficiency?**
>
> We agree that a fair efficiency comparison requires matched configurations, and we **have indeed ensured this** throughout the evaluation.
>
> * **Prompt Design**: As detailed in Supplementary Material C.2 (Prompt Designs), we intentionally kept the prompt nearly identical to that of VisionReasoner, adding only a single instruction: *"the explicit referring description for object localization"*. As shown in Figure 5 of Supplementary Material, the generated descriptions are typically 1–3 words, so the additional token overhead is minimal.
>
> * **Training Efficiency**: We have already provided an analysis in Supplementary Material D.1 (Analysis of Training Efficiency). Although DR²Seg employs two-stage rollouts during training, the total generated token length is significantly reduced, meaning the training time remains comparable. VisionReasoner takes about 3 hours 15 minutes, while DR²Seg completes in around 3 hours 10 minutes.
>
> * **Inference Efficiency**: In response to your suggestion, we report the wall-clock inference time under matched configurations (batch size = 1) and the same GPU (L40). On average:
>
>   * VisionReasoner: 4.43 s/sample
>   * DR²Seg: 3.67 s/sample
>
>    This represents a \~20% improvement in inference speed.
>
> These results collectively demonstrate that DR²Seg achieves superior efficiency, while achieving notable accuracy improvements.

---

> > ### Author Rebuttal · Reviewer_qLaq · 2026-04-05
> >
> > Thanks for careful response. I have no  any further questions.

---

> > > ### Author Response · Authors · 2026-04-06
> > >
> > > Dear  Reviewer qLaq:
> > >
> > > Thank you very much for your time and for acknowledging our efforts during the rebuttal. We are very glad to hear that our responses have well addressed your concerns.
> > >
> > > Sincerely,
> > >
> > > Authors of Submission 3042

---

### Decision · Program_Chairs · 2026-04-30

**Decision:**

Accept (regular)

**Comment:**

This paper proposes DR2Seg, a two-stage self-rewarding RL framework for reasoning segmentation in multimodal LLMs, which first generates an explicit target description and then performs referring segmentation. The method is motivated by reducing “overthinking” and is validated with strong results on ReasonSeg and RefCOCO. Reviewers generally found the paper well motivated, and highlighting the importance of the problem and the effectiveness of the two-stage design. The main concerns focused on novelty relative to prior self-rewarding approaches, limited theoretical grounding, insufficient failure-case analysis, efficiency comparisons, and evaluation in more complex settings. In response, the authors clarified the distinction from prior work and provided additional empirical evidence on failure cases, matched efficiency, the usefulness of the generated description, and broader settings, which resolved or substantially reduced most concerns. Overall, this is a solid empirical contribution with clear practical value. Therefore, the AC recommends acceptance of this submission.